# Dynamic evolution of urban infrastructure resilience and its spatial spillover effects: An empirical study from China

Hao Wang[1], Zhiying Huang[2]*, Yanqing Liang [1,3,4,5]*, Qingxi Zhang[1], Shaoxiong Hu[6], Liye Cui[7], Xiangyun An[1]

1 School of Geographic Sciences, Hebei Normal University, Shijiazhuang, China, 2 School of Land Science and Spatial Planning, Hebei University of Geosciences, Shijiazhuang, China, 3 Hebei Key Laboratory of Environmental Change and Ecological Construction, Shijiazhuang, China, 4 Hebei Technology Innovation Center for Remote Sensing Identification of Environmental Changes, Shijiazhuang, China, 5 Hebei Key Research Institute of Humanities and Social Sciences at Universities "Geo Computation and Planning Center of Hebei Normal University", Shijiazhuang, China, 6 School of Tourism and Environment, Zhangjiakou University, Zhangjiakou, China, 7 National Territory Spatial Planning Research Center of Hebei Province, Shijiazhuang, China

* hzy366@sohu.com (ZH); liangyanqing@126.com (YL)

**Data Availability Statement:** All relevant data are within the paper and its Supporting information files.

## Abstract

Urban infrastructure resilience is an important perspective for measuring the development quality of resilient cities and an important way to measure the level of infrastructure development. This paper uses the kernel density estimation, exploratory spatial data analysis, and spatial econometric models to analyze the characteristics of dynamic evolution and the spillover effects of the infrastructure resilience levels in 283 prefecture-level and above cities in China from 2010 to 2019. Our results are as follows. (1) The overall level of urban infrastructure resilience increased. The eastern region had a higher level than the national average. In contrast, the central, western and north-eastern regions had a slightly lower level than the national average. (2) The areas with high and higher resilience levels were mostly cities with more developed economic and social conditions in Eastern China. The areas below moderate resilience levels show a certain degree of clustering and mainly include some cities in Central, Western, and Northeast China. (3) The national level of urban infrastructure resilience shows significant spatial clustering characteristics, and the spatial pattern from coastal to inland regions presents a hotspot-subhotspot-subcoldspot-coldspot distribution. (4) There is a differential spatial spillover effect of national urban infrastructure resilience, which is gradually strengthened under the role of the economy, financial development, population agglomeration and government funding and weakened under the role of urbanization, market consumption and infrastructure investment. By exploring the dynamic evolution of infrastructure resilience in cities at the prefecture level and above and its spatial spillover effects, we provide a scientific basis for avoiding the siphoning effect among cities, improving the level of infrastructure resilience, and guiding the construction and development of resilient cities.

**Funding:** This research was funded by National Natural Science Foundation of China (41471090), Yanqing Liang. Key Projects of Science and Technology Research of Hebei Provincial Education Department (ZD2019115), Yanqing Liang. Funders of this study, Prof. Yanqing Liang, took on the roles of conceptualization, data curation, software, formal analysis, funding acquisition, project administration, resources, writing-review and editing.

**Competing interests:** The authors have declared that no competing interests exist.

## Introduction

As socio-ecological systems, cities are not only complex but also face serious acute shocks and chronic pressure challenges [1]. As the lifeline project of cities, urban infrastructure is not only an important material carrier for population concentration and the material basis for urban economic and social development but also an important symbol of urban civilization and modernization [2]. Currently, urban infrastructure in China exists in multiple disturbing environments that include sudden public health events, the impacts of natural disasters and ageing facilities, forcing cities to cope with emergencies and resist disasters with greater security risks. Urban infrastructure resilience (UIR) is the ability of infrastructure to cope with and recover from disturbances. It is an important indicator for measuring the level of infrastructure development and an important perspective for measuring the quality of resilient city development. Additionally, it can reflect the strength of infrastructure resilience, and improving the level of UIR plays an important role in improving urban governance and promoting resilient city development. Studies assessing the composition and classification of urban infrastructure have all classified it into six major systems and their combinations [3–13]: water supply and drainage, energy supply, road transport, postal and telecommunications, landscaping and sanitation, and disaster prevention facilities. However, these studies have not clearly described internal and external boundaries, and only the transport facilities system is divided between inner and outer urban transport facilities. We consider the internal and external boundaries of urban infrastructure as the use of infrastructure and services outside the administrative boundaries of the city. Thus, we define the research object of this paper as the infrastructure of energy, supply and drainage, transportation, postal, and telecommunications within the city, with the boundary of administrative districts.

Resilience was originally used in mechanics and physics to represent the ability of an object to return to its original shape after being deformed by an external force. In 1973, Canadian ecologist Holling first combined the theory of resilience with the discipline of ecology to describe "the ability of an ecosystem to return to a stable state after a disturbance" [14]. As the times have evolved, resilience has become more closely integrated with various disciplines in the field, thus giving it a richer meaning. The concept and theory of resilience has evolved from "equilibrium" to "adaptation" and from "single ecosystems" to "social-ecological complex systems". For the connotation of UIR, most scholars explain it in terms of the three characteristics of resilience (resistance, absorption, and recovery). Bruneau et al. considered it to be the ability of urban infrastructure to cope and recover from disasters [15]. Haimes et al. considered infrastructure resilience as the ability of an infrastructure system to resist external disruptions at a tolerable cost and to recover within an acceptable cost and time frame [16]. Gay et al. defined it as the ability of civil infrastructure systems to reduce the loss of performance in the case of disturbances and to recover to a certain level of performance within acceptable time and cost constraints [17]. In summary, UIR can be understood as the ability of infrastructure to withstand disturbances, absorb losses, and promptly return to normal operating conditions.

Research on UIR focuses on the evolution of spatial and temporal patterns, influencing factors, dynamic modelling, system external coupling, and convergence. Zhang and Zhu et al. used the ESDA method to analyze the evolutionary characteristics of the spatial and temporal patterns of infrastructure resilience in Shandong Province and the three major urban agglomerations in China, respectively, and concluded that rapid urbanization would put pressure on the infrastructure resilience of some cities in Shandong Province, with the overall gap showing a tendency to increase before decreasing, while the level of infrastructure resilience in the three major urban agglomerations in China was generally on the rise [18, 19]. Based on the analysis of the evolutionary characteristics of the spatio-temporal pattern of UIR during the current

time period, Zhang and Chen et al. used BP neural networks to simulate dynamically the level of infrastructure resilience in the Beijing-Tianjin-Hebei and Harbin-Changchun urban agglomerations [20, 21]. Zhang et al. used a convergence model to validate the convergence and divergence trends of UIR in northeastern China [22]; Positive influences on the resilience of urban infrastructure include the level of economic development, administrative capacity and innovation [18, 19], while negative influences include the degree of openness and the level of science and technology [23]. The study of external coupling of UIR systems is also a hot topic, with Zhou and Bai et al. investigating the coupling and coordination of infrastructure resilience systems with urbanization levels and land use efficiency, respectively [24, 25]. The scale of research on UIR covers the whole country [26–28], cities [6, 18, 23, 29, 30], and urban agglomerations [31–34], etc. The quantitative assessment methods mainly involve entropy weight method [18, 19, 23, 32, 33], hierarchical analysis [27, 28], and factor analysis [35], and system function curve methods [36–38], etc.

There is a wealth of research on UIR, and the advancement of previous theoretical research and the accumulation of empirical experience has provided a rich research base and a solid research foundation, but there are also aspects that can be further explored. First, previous studies have rarely taken the perspective of the connotations of infrastructure, thus making it difficult to reflect the comprehensiveness of infrastructure resilience systems. Second, previous studies have rarely considered the spatial spillover effects of external factors on the resilience of urban infrastructure. Third, previous studies have mostly evaluated UIR at the provincial, municipal and city cluster scales, ignoring the unevenness arising from the large differences in infrastructure resilience levels between cities across the country. In summary, first, this paper constructs a comprehensive UIR evaluation system by integrating energy, supply and drainage, transport, postal and communication, environmental, and cultural and health facilities. Second, this paper explores the spatial spillover effects and causes of the variability in infrastructure resilience among cities at different time points using a spatial econometric model that considers spatial dependence. Again, this paper takes 283 cities at the prefecture level and above in China as its research objects to more intuitively reflect the spatial heterogeneity of UIR levels at the municipal scale, which is an important socioeconomic management unit.

Based on the capacity assessment dimension of resilience theory [39], this study assesses the level of UIR from the perspective of the existing capacity of urban infrastructure subsystems and explores the dynamic evolution characteristics and spatial spillover effects of UIR. It helps to broaden the application field of resilience theory and expand the research horizon in the field of urban infrastructure, and it can provide references for other countries, particularly developing countries, to promote coordinated regional development, guide the construction of resilient cities, and reasonably avoid the urban siphon effect.

## Materials and methods

### Studied regions

Considering the consistency of the study units and the availability of data, this article takes 2019 prefecture-level and above cities as the standard. It identifies 283 cities at the prefecture level and above (hereinafter collectively referred to as "cities") as the research objects based on the 2019 administrative divisions (excluding Hong Kong, Macao, Taiwan and Tibet). Different levels of social and economic development among the cities lead to different levels of urbanization development and levels of UIR. The differences in the level of urbanization development and the level of UIR make it possible to identify the factors influencing UIR.

## Data sources and processing

The data were mainly obtained from the 2011, 2016, and 2020 China Urban Statistical Year-book and China Urban Construction Statistical Yearbook as well as provincial and municipal statistical yearbooks, and national economic and social development statistical bulletins, with some missing data calculated based on the exponential smoothing method. The vector data on China's administrative divisions were obtained from the National Geographic Information Centre (http://www.ngcc.cn/ngcc/). Due to the inconsistent units and the existence of positive and negative indicators for all of the index data, dimensionless processing was carried out based on the polarization standardization method, and the calculation formula is:

$$\text{For positive indicators}: X'_{\theta ij} = \frac{X_{\theta ij} - X_{\theta ijmin}}{X_{\theta ijmax} - X_{\theta ijmin}} \tag{1}$$

$$\text{For negative indicators}: X'_{\theta ij} = \frac{X_{\theta ijmax} - X_{\theta ij}}{X_{\theta ijmax} - X_{\theta ijmin}} \tag{2}$$

Where $X'_{\theta ij}$ is the standardized value; $X_{\theta ij}$ is the value of the $jth$ indicator for the city $i$ in year $\theta$; $X_{\theta ijmax}$ is the maximum value of the $jth$ indicator for the city $i$ in year $\theta$; $X_{\theta ijmin}$ is the minimum value of the $jth$ indicator value for the city $i$ in year $\theta$.

## Construction of the urban infrastructure resilience index system

The current index system for urban infrastructure and its resilience is mainly constructed in terms of energy, water supply and drainage, transportation, communication and other munici-pal utilities. For example, Chen et al. described the state of infrastructure resilience in terms of the gas penetration rate, water penetration rate, and road area per capita [6]. Liu, Chen, Tao, Bai, and Zhang et al. constructed indicator systems including the daily domestic water con-sumption per capita, water supply pipeline density, drainage pipeline density, and road area per capita [5, 7–13, 26, 27, 40–42].

This study follows the principles of comprehensiveness, importance, measurability, and availability of evaluation indicators and uses the "Standard for Basic Terminology of Urban Planning" (GB/T50280-98) as the basis for selection. The frequency of the indicators involved in 32 highly relevant papers is counted, and 21 indicators with representative and high usage are selected. Thus, we have selected four indicators, namely, special vehicles for sanitation for 10000 people, the number of secondary schools, the number of elementary schools, and library collections per 100 people, as useful supplements to the engineering facilities and social service facilities in the connotation of infrastructure together to ensure that the index system is more comprehensive. The finalized UIR evaluation index system consists of 25 indicators such as the gas penetration rate, total gas supply, total electricity consumption, and water supply pipe-line density (Table 1).

## Research methods

**Mean squared error decision method.**   The mean squared difference decision method is an objective weighting method. The basic principle of this method is that the coefficient of each indicator weight depends on the relative dispersion of the indicator attribute values. Additionally, the greater the dispersion is, the greater the indicator weight coefficient, which has the advantage of high weighting accuracy [43]. In this paper, we use this method to deter-mine the weight of each indicator of UIR and apply the composite index method to calculate the composite score of UIR.

**Table 1. Comprehensive evaluation index system of urban infrastructure resilience.**

| Sub-index | Index | Property | Frequency | Weight |
|---|---|---|---|---|
| Resilience of energy facilities | Gas penetration rate | Positive | 32 | 0.043 |
| | Total gas supply | Positive | 28 | 0.026 |
| | Total electricity consumption | Positive | 25 | 0.040 |
| Resilience of supply and drainage facilities | Water supply pipeline density | Positive | 32 | 0.057 |
| | Water penetration rate | Positive | 28 | 0.037 |
| | Daily domestic water consumption per capita | Negative | 30 | 0.043 |
| | Drainage pipeline density | Positive | 32 | 0.047 |
| Resilience of transport facilities | Road area per capita | Positive | 32 | 0.048 |
| | Road network density | Positive | 29 | 0.047 |
| | Number of buses per 10,000 people | Positive | 31 | 0.032 |
| | Total number of bus passengers | Positive | 28 | 0.039 |
| Resilience of postal and communication facilities | Number of households with internet access per 10,000 people | Positive | 30 | 0.061 |
| | Number of households with cell phones per 10,000 people | Positive | 32 | 0.033 |
| | Telecommunications services per capita | Positive | 29 | 0.035 |
| | Postal business per capita | Positive | 28 | 0.036 |
| Resilience of environmental facilities | Centralized urban sewage treatment rate | Positive | 25 | 0.036 |
| | Harmless disposal rate of domestic waste | Positive | 26 | 0.020 |
| | Special vehicles for sanitation for 10,000 people | Positive | 10 | 0.030 |
| | Green space per capita | Positive | 29 | 0.047 |
| | Greening coverage of built-up areas | Positive | 25 | 0.044 |
| Resilience of cultural and health facilities | Number of secondary schools | Positive | 5 | 0.040 |
| | Number of primary schools | Positive | 4 | 0.049 |
| | Library collections per 100 people | Positive | 10 | 0.042 |
| | Number of hospital beds per 1,000 population | Positive | 26 | 0.036 |
| | Number of health institutions | Positive | 29 | 0.032 |

Note: The positive indicator means that the higher the value, the higher the level of infrastructure resilience. The negative indicator means that the higher the value, the lower the level of infrastructure resilience.

1. Calculate the mean value of the random variable $X$ $E(X_j)$

$$E(Xj) = \frac{1}{n}\sum_{i=1}^{n} Xij \tag{3}$$

2. Calculate the mean squared deviation of indicator $X_j$ $F(X_j)$

$$F(Xj) = \sqrt{\sum_{i=1}^{n} (Xij - E(Xj))^2} \tag{4}$$

3. Calculate the weight of indicator $X_j$ $W(X_j)$

$$W\left(X_j\right) = \frac{F\left(X_j\right)}{\sum_{i=1}^{n} F\left(X_j\right)} \tag{5}$$

4. The composite index method is used to calculate the composite score of UIR, and the calculation formula is:

$$IR_i = \sum X'_{\theta ij} W_j \tag{6}$$

Where $W_j$ is the weight of each indicator; $IR_i$ is the composite score of infrastructure resilience of city $i$.

**Kernel density estimation.** Kernel densities are often used in probability theory to estimate unknown density functions [44]. Kernel estimation is a nonparametric estimation method that is mainly used for estimating the density functions of random variables [45–47]. This paper analyzes the shape, two-tailed extension, and kurtosis of the kernel density estimation curve to reveal the spatio-temporal evolutionary trend of the level of UIR. The calculation formula is:

$$f(IR) = \frac{1}{nh} \sum_{i}^{n} k \left( \frac{IR - IR_i}{h} \right) \tag{7}$$

Where $f(IR)$ is the city infrastructure resilience kernel density function, $n$ is the number of cities, $h$ is the bandwidth controlling the smoothness of the kernel density curve, $IR_i$ is the infrastructure resilience value of city $i$, and $k$ () denotes the kernel function.

**Exploratory Spatial Data Analysis (ESDA).** ESDA is centered on spatial correlation measurement, and it reveals the spatial dependencies among research objects by describing and visualizing the spatial distribution pattern of things or phenomena [48]. The method includes global and local spatial autocorrelation analysis, allowing it to clarify the spatial distribution characteristics of the level of UIR nationwide and detect its influence on surrounding spatial units. Global spatial autocorrelation is used to analyze the overall spatial correlation pattern of observations in geographic space, and it is usually denoted by the global Moran's $I$. Local spatial autocorrelation is used to further analyze the spatial correlation of observations in urban areas. Local spatial autocorrelation is used to further verify the existence of spatial clustering of high and low values, and it is mainly measured by the Getis-Ord $G^*i$-index and can characterize the spatial differentiation of the study units in a region in more detail. The calculation formula is:

$$Global\ Moran's\ I = \frac{n \sum_{i=1}^{n} \sum_{j=1}^{n} W_{ij} \left( IR_i - \overline{IR} \right) \left( IR_j - \overline{IR} \right)}{\sum_{i=1}^{n} \sum_{j=1}^{n} W_{ij} \sum_{i=1}^{n} \left( IR_i - \overline{IR} \right)^2} \tag{8}$$

$$G_i^*(d) = \frac{\sum_{j=1}^{n} W_{ij}(d) IR_j}{\sum_{j=1}^{n} IR_j} \tag{9}$$

Where $n$ is the number of study areas; $IR_i$ and $IR_j$ are the observed values of spatial units, respectively; $\overline{IR}$ is the mean value of the observed values; $W_{ij}$ is the spatial weight matrix.

**Spatial econometric model.** To better explain the spatial dependence effect of UIR, regression models from spatial econometrics, commonly known as spatial lag model (SLM)

and spatial error model (SEM) [49], are used for fitting. Among them, SLM (Eq 10) incorporates a spatial lag term of UIR to characterize the dependence of the infrastructure resilience of a city on the infrastructure resilience of neighboring cities. SEM (Eqs 11 and 12) deals with the spatial dependence effect by incorporating the spatial autocorrelation of the error term.

$$Y = \rho WY + X\beta + \varepsilon \tag{10}$$

$$Y = X\beta + \varepsilon \tag{11}$$

$$\varepsilon = \lambda W\varepsilon + \mu \tag{12}$$

Where the dependent variable $Y$ is a vector of $n \times 1$; the independent variable $X$ is a data matrix of $n \times k$; the regression coefficient $\beta$ is a vector of $k \times 1$; the random error term $\varepsilon$ is a vector of $n \times 1$; $W$ is a spatial weight matrix, where a spatial geographic distance matrix is used; $\rho$ is the coefficient of the endogenous interaction effect ($WY$), whose magnitude reflects the degree of spatial spillover; $\mu$ is a vector of normally distributed random errors; $\lambda$ is the spatial correlation coefficient between regression residuals.

## Results

### Dynamic evolution of urban infrastructure resilience

**Characteristics of the temporal evolution of urban infrastructure resilience.** Through comprehensive calculations, the levels of UIR of the country and its four major regions from 2010 to 2019 can be derived (Fig 1). The national level of UIR rose during the study period from 0.282 to 0.324. For the eastern region, the level rose from 0.322 to 0.360, which is much

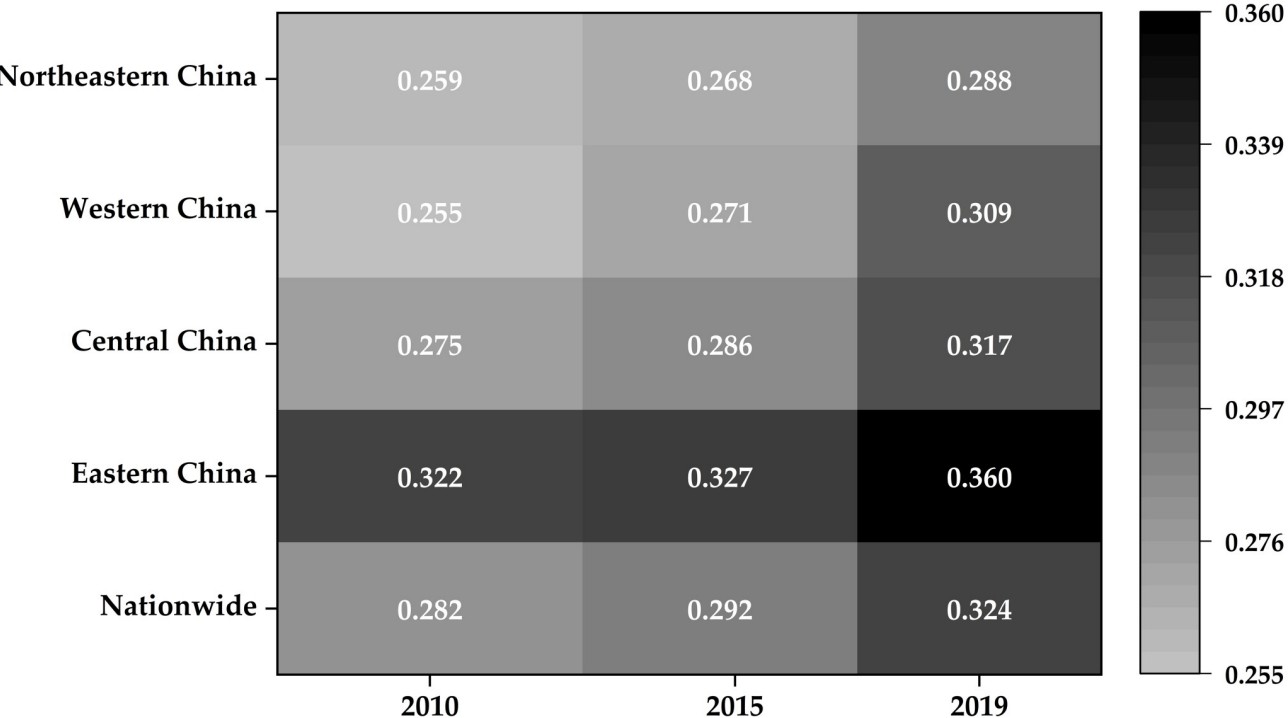

**Fig 1. National urban infrastructure resilience comprehensive score.**

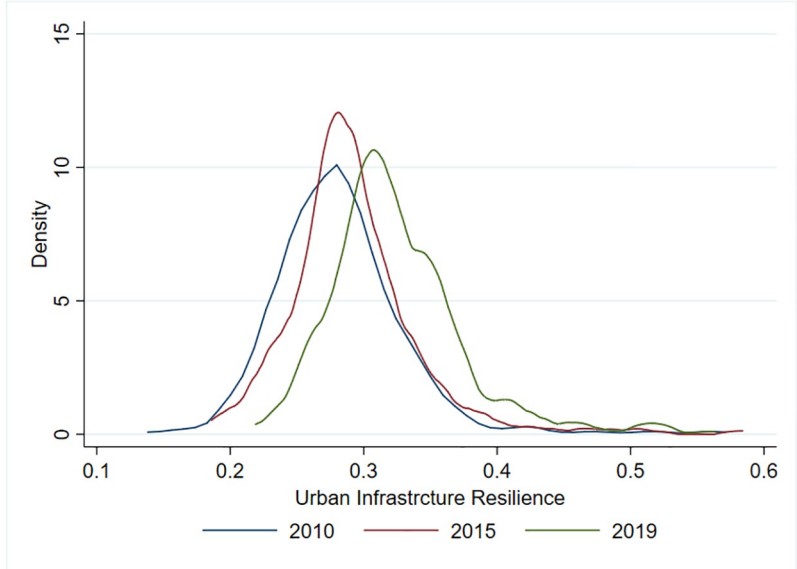

**Fig 2. Kernel density estimates of national urban infrastructure resilience levels.**

higher than the national level and the levels of the central, western, and north-eastern regions. In contrast, the levels of the central and western regions were slightly lower than the national level. The north-eastern region had the lowest level.

The kernel density estimation of the levels of UIR in 2010, 2015, and 2019 was conducted using Stata 15.1 software to reveal the characteristics of the evolution of the distribution of UIR across China (Fig 2). From 2010 to 2019, the overall trend of the skewed distribution of the levels of UIR across the country changed. In general, the kernel density curve shifted to the right, showing a single-peak-bi-peak evolutionary trajectory. Additionally, the peaks became steeper in general, indicating that the overall level of UIR improved and that the number of areas with lower resilience levels decreased but that the regional differences increased. The nuclear density curves in 2010 and 2015 were single-peaked structures, and the peaks were approximately 0.27, with the peaks on the left and the trailing phenomenon on the right, indicating that the scores are discrete. Although there were cities with high resilience levels, most of them were scattered in the low and lower resilience levels, with large disparities between cities. The nuclear density curve had a bi-peak structure in 2019, indicating a clear polarization of the national level of UIR, with the highest height of the main peak and a sharp peak. These results for 2019 indicate the highest convergence of the score data compared with 2010 and 2015, but the main peak was approximately 0.31, indicating that the vast majority of cities were at a moderate level of infrastructure resilience in 2019. The second peak was approximately 0.52, indicating that there were significantly more cities with high levels of resilience in 2019 than in the other years.

**Characteristics of the spatial evolution of urban infrastructure resilience.** Through comprehensive calculation and the use of ArcGIS10.2 software, the national level of UIR is divided into five levels: higher, high, moderate, low, and lower resilience levels (Table 2, Fig 3). In 2010, the areas with a higher resilience level included only Beijing, Shanghai, and Shenzhen, which were included in the three major city clusters of Beijing-Tianjin-Hebei, Yangtze River Delta and Pearl River Delta. There were 14 cities with a high level of resilience, including

**Table 2. Urban infrastructure resilience classification.**

| Grades of Urban Infrastructure Resilience | Grade I | Grade II | Grade III | Grade IV | Grade V |
|---|---|---|---|---|---|
| | Lower Resilience | Low Resilience | Moderate Resilience | High Resilience | Higher Resilience |
| Comprehensive Evaluation Value of Urban Infrastructure Resilience | <0.200 | 0.200–0.300 | 0.300–0.350 | 0.350–0.450 | >0.450 |

Chongqing and some cities on the eastern coast. The areas with a moderate resilience level were mostly distributed around larger cities. The areas with a lower resilience level mainly included Chongzuo, Anshun, Ankang, Baiyin, and Longnan. In 2015, the gap in the level of infrastructure resilience of cities across the country expanded to a certain extent, with 2 cities increased, Zhuhai and Dongguan, becoming cities with a higher resilience level compared to 2010. Additionally, the number of cities with high resilience level increased to 19, mostly located on the eastern coast as well as municipalities directly under the central government and provincial capitals, basically showing a trend of gradually decreasing from east to west. Suzhou, Guangzhou, Chongqing and Jiayuguan city were added to the high resilience level areas in 2019. The areas with a high and moderate resilience level show a trend of gradually expanding from east to center. The areas with a low resilience level were distributed in Shanxi, Inner Mongolia, Liaoning, Jilin, Heilongjiang, and Anhui Provinces. There were no areas with a lower resilience level. This result indicates that with the continuous development of the urban economy and increasing investment in infrastructure construction, the level of UIR improved.

Overall, the evolution of the spatial pattern of the national level of UIR is mainly reflected in the following aspects: (1) Cities above a high resilience level are basically cities with more developed economic and social conditions, among which Beijing, Shanghai, and Shenzhen are always high-level areas, but with economic development showing a trend of expansion from the coastal to inland regions. (2) Cities above the high resilience level are basically cities with more developed economies in coastal regions as well as provincial capitals and sub-provincial cities, with an overall trend of spreading from coastal regions to inland regions. (3) Cities

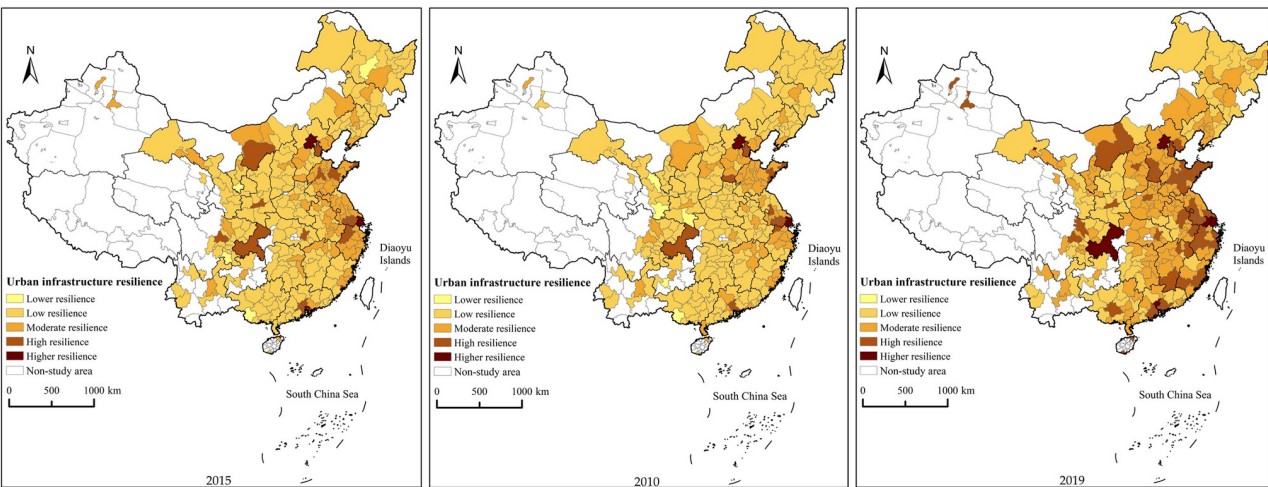

**Fig 3. Spatio-temporal pattern of resilience level of urban infrastructure in China from 2010 to 2019.** (The map was prepared in ArcGIS 10.1 using political boundaries from the National Geomatics Center of China (http://www.ngcc.cn/ngcc/)).

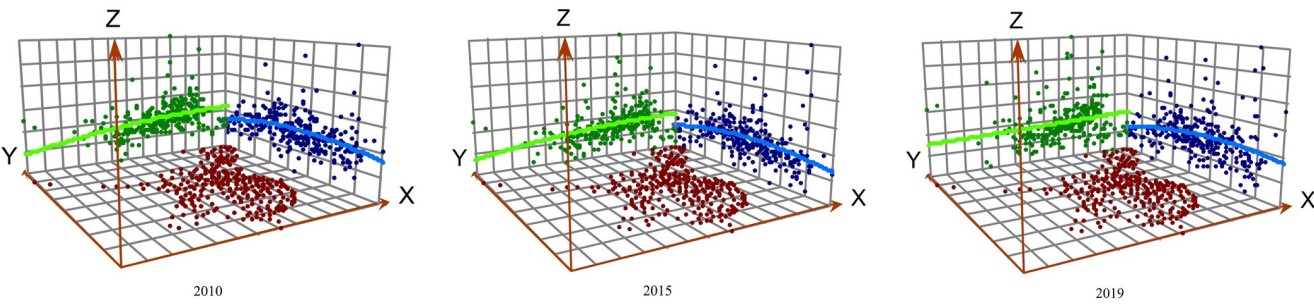

**Fig 4. Trend analysis of urban infrastructure resilience in China from 2010 to 2019.**

below the moderate resilience level show a certain degree of agglomeration. These are mainly some cities in Central China and most cities in the west and northeast. Additionally, the overall level of infrastructure resilience of these cities is not optimistic compared with economically developed cities.

With social and economic development, the level of UIR across the country has been spatially changing. By fitting discrete point data to a continuous surface through the geostatistical module, a trend map of the national level of UIR is obtained (Fig 4). This map reflects the spatial differences in the level of UIR, where the green trend line represents the east–west directional change and the blue trend line represents the north–south directional change. The figure illustrates that the national level of UIR from 2010 to 2019 shows a distribution trend of gradually decreasing from east to west and gradually increasing from north to south. These results indicate the existence of significant regional incongruities.

**Spatial clustering characteristics of urban infrastructure resilience.** Based on Arc-GIS10.2 software, the global Moran's *I* of national UIR in 2010, 2015, and 2019 was calculated, as shown in Table 3. The results show that the global Moran's *I* is greater than 0 and shows a trend of rising and then falling, from 0.3840 to 0.4032 and to 0.3816, with z values greater than 2.58 (the critical value at a 1% significance level) and p values of 0. Based on these results, national UIR shows significant spatial clustering and positive spatial correlation characteristics during the period. The spatial distribution of cities with similar levels of infrastructure resilience is more concentrated.

To further explore the spatial clustering of the level of the UIR across China, the results of the local spatial correlation Getis-Ord G* index were classified using ArcGIS 10.2 software, and the study area was divided into hotspot, subhotspot, subcoldspot, and coldspot areas (Fig 5).

In 2010, the hotspot areas were mainly concentrated in the Beijing-Tianjin-Hebei, Yangtze River Delta, Pearl River Delta, and Shandong Peninsula urban clusters. The cities in these areas and neighboring cities had high infrastructure resilience levels, which were positively correlated and spatially formed high-value clusters. The cold-spot areas were mainly distributed in northern Heilongjiang, central and south-eastern Gansu, north-eastern and western

**Table 3. Changes of urban infrastructure resilience global Moran's *I* index.**

| Year | 2010 | 2015 | 2019 |
|---|---|---|---|
| Moran's *I* | 0.3840 | 0.4032 | 0.3816 |
| *z-value* | 9.7580 | 10.2534 | 9.6566 |
| *P-value* | 0.0000 | 0.0000 | 0.0000 |

Guangxi, western Guizhou, and the Cheng-du-Chongqing urban agglomeration. The infrastructure resilience level of these cities and the neighboring cities in these areas was low and positively correlated, forming low-value spatial agglomeration. The subhotspot areas were distributed around the hotspot areas. The subcoldspot areas were mainly distributed around the coldspot areas, including north-eastern Heilongjiang, Gansu, Shaanxi, Sichuan and some cities in Guangxi. There was no obvious spatial clustering point in this area. In 2015, the location of the hotspot areas did not change, but the area decreased. The area of the cold-spot areas also decreased, mainly in eastern and northern Heilongjiang, eastern Gansu, Ningxia and western Yunnan, Shaanxi, and southern Henan. The subcoldspot areas were distributed in Heilongjiang, Henan, Shaanxi, Sichuan, eastern Gansu and Guangxi. The location of the subhotspot areas did not change, but the area decreased. By 2019, the hotspot areas included the Beijing-Tianjin-Hebei, Yangtze River Delta, and Pearl River Delta city clusters. The area of the hotspot areas of the Beijing-Tianjin-Hebei and Pearl River Delta city clusters decreased, and the area of the hotspot areas of the Yangtze River Delta city cluster increased. The coldspot areas were distributed in some cities in the northeast region, Gansu, Shaanxi, Sichuan, Yunnan, and Guangdong. The subhotspot areas were distributed in Hulunbuir and some cities in Northeast China, Shaanxi, Henan Sichuan, Yunnan, and Guangxi, in addition to some other cities.

In general, the hotspot areas of UIR are distributed in the Beijing-Tianjin-Hebei, Yangtze River Delta, Pearl River Delta, and Shandong Peninsula urban clusters, while the coldspot areas are gradually decreasing in size and are mostly located in the northeast and central regions. The spatial pattern from coastal to inland regions shows a hotspot-subhotspot-sub-coldspot-coldspot distribution. The distribution pattern of coldspots and hotspots shows that there is a significant positive correlation between the level of UIR and the level of economic development.

## Spatial spillover effects of urban infrastructure resilience

The non-equilibrium state of economic, social, and resource factors in cities across China and the spatial interaction effects among neighboring regions jointly contribute to the formation of and changes in the spatial patterns of UIR. Is there a significant spillover effect of infrastructure resilience among cities? Are there any differences in the contributions of each factor to the spillover effect? To address these questions, drawing on relevant studies by established scholars of resilient cities [18, 19, 23], the infrastructure development level [50–53], and the influencing factors of urban development quality [54–56] and based on the principles of scientificity, rationality, and accessibility, we select population density (X1), the economic development level (X2), the urbanization level (X3), the industrial structure (X4), the financial development level (X5), the market consumption level (X6), the infrastructure investment level (X7), and the local government financial expenditure level (X8) as explanatory variables; they are measured by the total population/administrative land area, GDP per capita, the urbanization rate, the proportion of the value added of the tertiary industry in GDP, the proportion of the loan balance of local financial institutions in GDP, the total retail sales of consumer goods per capita, the completed infrastructure investment in the year, and total government financial expenditure, respectively. The spillover effects of UIR in 2010, 2015, and 2019 are verified and analyzed based on traditional ordinary least squares (OLS) linear regression models, SLMs, and SEMs.

**Spatial econometric model testing.** From the above global Moran's *I* and coldspot and hotspot map, it is clear that there is a significant spatial correlation in the level of UIR. As shown in Tables 4 and 5, the residual Moran's *I* statistics for 2010, 2015, and 2019 are significant at the 1% level, which also indicates that there is a significant spatial dependence in the

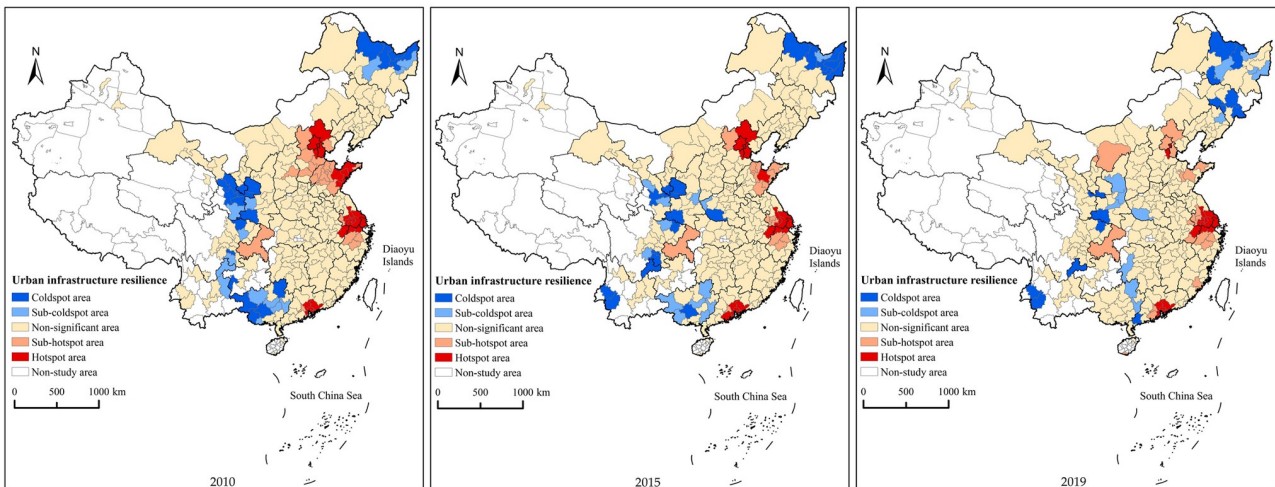

**Fig 5. Analysis of cold and hot spots of urban infrastructure resilience in China from 2010 to 2019.** (The map was prepared in ArcGIS 10.1 using political boundaries from the National Geomatics Center of China (http://www.ngcc.cn/ngcc/)).

OLS model residuals. Therefore, a spatial regression model that takes spatial dependence into account is needed to analyze the spillover effects of UIR in China. By comparing the Lagrange multiplier (LM) test and its robust form (R-LM test), it can be seen that in 2010, LMerr was significant at the 1% level, R-LMerr was significant at the 10% level, and both LMlag and R-LMlag were significant at the 1% level. In 2015, LMerr, R-LMerr, LMlag, and R-LMlag were all significant at the 1% level. The statistics for both LMlag and R-LMlag were greater than the statistics for LMerr and R-LMerr. In 2019, LMerr and R-LMerr were significant at the 1% and 5% levels, respectively, while both LMlag and R-LMlag were significant at the 1% level. The statistics for LMlag and R-LMlag were greater than the statistics for LMerr and R-LMerr. Based on the model selection criterion given by Anselin [57], the SLM is an appropriate spatial econometric model for 2010, 2015, and 2019. Meanwhile, based on the R values in Table 5, the R values of the SLM for all three years are greater than those of the OLS model and SEM, indicating that it is reasonable to choose the SLM for spatial analysis.

**Spatial lag model regression results.** The estimation results of the SLM are shown in Table 6, and the coefficients of the spatial error terms for 2010, 2015, and 2019 are 0.835, 0.884, and 0.605, respectively. All of these coefficients are significant at the 1% level. These results indicate that there is a positive spillover effect of national UIR and that an increase in the level of infrastructure resilience in one city will lead to an increase in the level of infrastructure resilience in neighboring cities. Additionally, a convergence effect between neighboring

**Table 4. Spatial dependence test.**

| Year | Inspection | Residuals Moran's *I* | LMerr | R-LMerr | LMlag | R-LMlag |
|------|-----------|----------------------|-------|---------|-------|---------|
| 2010 | Statistics | 31.174 | 55.282 | 43.555 | 44.714 | 32.987 |
| | Probability value | 0.000 | 0.000 | 0.070 | 0.000 | 0.000 |
| 2015 | Statistics | 20.328 | 22.617 | 14.040 | 47.612 | 39.034 |
| | Probability value | 0.000 | 0.000 | 0.000 | 0.000 | 0.000 |
| 2019 | Statistics | 12.217 | 7.727 | 5.281 | 11.624 | 9.178 |
| | Probability value | 0.000 | 0.005 | 0.022 | 0.001 | 0.002 |

**Table 5. OLS, SEM and SLM model R values.**

| Year | OLS | SEM | SLM |
|------|-----|-----|-----|
| 2010 | 0.631 | 0.631 | 0.689 |
| 2015 | 0.714 | 0.716 | 0.759 |
| 2019 | 0.707 | 0.714 | 0.729 |

regions is formed. The regression coefficients and significance levels of each explanatory variable varied significantly during the study period.

The regression coefficients of GDP per capita (X3) and government financial expenditure (X8) in 2010 were significantly positive at the 1% level, 0.410 and 0.251, respectively, with the former contributing the most significant spillover effect on the level of infrastructure resilience of neighboring cities. GDP per capita is an indicator of economic development in developed economies. Economic development is the material source of infrastructure supply formation, and the level of economic development is conducive to promoting growth in infrastructure demand. An increase in areas with high economic development levels will not only promote the large-scale construction and development of local infrastructure but also drive the infrastructure development of neighboring regions, thus promoting an increase in the level of infrastructure resilience of local and neighboring cities. An increase in government fiscal spending can create a stronger driving force for infrastructure resilience, increasing the level of infrastructure resilience by stimulating market efficiency and dynamism, improving capital allocation efficiency, and bringing advanced technology and management experience. The regression coefficients of infrastructure investment (X7) and the industrial structure (X4) are significantly positive at the 5% level, 0.123 and 0.095, respectively. Regions with higher infrastructure investment indirectly promote economic development and drive neighboring cities to carry out infrastructure construction through the diffusion effect, thus promoting an improvement in infrastructure resilience levels in local and neighboring regions. The optimization and upgrading of the industrial structure drive high-quality economic development, which indirectly promotes infrastructure development.

**Table 6. Spatial lag model estimation results.**

| Variables | 2010 | 2015 | 2019 |
|-----------|------|------|------|
| X1 | 0.039 | 0.018 | 0.086 |
| X2 | 0.410***(0.000) | 0.455***(0.000) | 0.791***(0.000) |
| X3 | 0.010 | -0.047 | -0.180**(0.031) |
| X4 | 0.095**(0.032) | 0.062*(0.086) | 0.113***(0.006) |
| X5 | -0.026 | 0.052 | 0.134**(0.028) |
| X6 | 0.166 | 0.188**(0.038) | 0.030 |
| X7 | 0.123**(0.044) | -0.033 | -0.009 |
| X8 | 0.251***(0.001) | 0.381***(0.000) | 0.267***(0.001) |
| $\lambda$ | 0.835***(0.000) | 0.884***(0.000) | 0.605***(0.000) |

Note: P-value in parentheses.

* significant at the 10% level.

** significant at the 5% level.

*** significant at the 1% level.

The regression coefficients of GDP per capita (X3) and government financial expenditure (X8) in 2015 are significantly positive at the 1% level, and both increase to 0.455 and 0.381, respectively. The former has the largest contribution, and the higher its value is, the more significant the spillover effect on the level of infrastructure resilience in neighboring cities. The regression coefficient of the total retail sales of consumer goods per capita (X6), 0.188, is significantly positive at the 5% level. As the starting and ending point of economic activities, national consumption has a guiding and pulling effect on economic growth and helps promote the transformation of economic growth momentum to realize the upgrading of economic quality and efficiency. Economic growth and upgrading not only drive local infrastructure development but also have a diffusion effect on infrastructure construction in neighboring regions, thus promoting an increase in infrastructure resilience in local and neighboring cities. The regression coefficient of the industrial structure (X4), 0.062, is significantly positive at the 10% level. As the driving force of economic development, the industrial structure directly drives the local economy and indirectly drives the improvement in infrastructure resilience levels in neighboring areas.

The regression coefficients of GDP per capita (X2), government financial expenditure (X8), and the industrial structure (X4) in 2019 are significantly positive at the 1% level, 0.791, 0.267, and 0.113, respectively. Additionally, the level of financial development (X5) changes from non-significant to significantly positive at the 5% level. These results show that economic development, financial expenditure capacity at the government level, industrial structure optimization and local financial development will not only promote the construction of infrastructure in a region but also drive the development of and improvement in infrastructure in neighboring regions, thus promoting an increase in the infrastructure resilience level in local and neighboring cities. The regression coefficient of the urbanization rate (X3) is -0.180 and significantly negative at the 5% level, and it has a significant negative spillover effect on UIR. Rapid urbanization places enormous pressure on infrastructure. Disorderly population expansion greatly exceeds the carrying capacity of urban infrastructure, also placing enormous pressure on infrastructures such as transportation, the environment, and public services. Additionally, the increase in the cost of land, labour, environmental protection, and financing incurs a high cost of infrastructure construction. Furthermore, the negative impact of rapid urbanization has a radiation effect on neighboring cities, which is not conducive to improving their infrastructure resilience.

The changes in the regression coefficients of the influencing factors from 2010 to 2019 show that the contributions of population density (X1), the economic development level (X2), the industrial structure (X4), the financial development level (X5), and the local government financial expenditure level (X8) to the growth in the national level of UIR are positive and fluctuate upwards. To some extent, these results reflect the spatial spillover effect of infrastructure resilience brought by the proximity of the economy, production diffusion, and labour sharing, which in turn reflects to some extent the increasing role of population concentration, economic development, and government funding. The contributions of the urbanization rate (X3), the total retail sales of consumer goods per capita (X6), and infrastructure investment (X7) to the growth in the national level of UIR is decreasing, indicating that the direct influence of the urbanization process, the consumer market, and infrastructure investment on local infrastructure resilience is greater than the indirect influence on neighboring cities, thus weakening the propagation of UIR spillover signals. The direction of urbanization should be grasped scientifically and precisely, the consumer market should be deployed reasonably, and private capital should be fully introduced to promote the development of resilient city construction.

## Conclusions and suggestions

### Conclusions

This paper explored the dynamic evolution of the spatial and temporal characteristics of UIR using kernel density estimation and ESDA methods for 283 prefecture-level and above cities in China. The optimal spatial econometric model was selected to explore its spatial spillover effects.

Our results showed that the level of UIR in China tended to increase over the study period. The eastern region had the highest UIR. Regions above the high resilience level showed a tendency to spread from coastal to inland areas, and regions below the moderate resilience level showed a degree of agglomeration. Cities with similar levels of infrastructure resilience exhibited a more concentrated spatial distribution, and the spatial distribution pattern from coastal to inland areas was a hotspot-subhotspot-subcoldspot-coldspot pattern.

In terms of the spatial spillover effects of UIR, different spatial dependencies on infrastructure resilience among cities were observed, and the spillover effects were prominent. Various factors had different degrees of influence on the spillover effects of UIR, and the spillover effects were mainly influenced by the population concentration, financial and economic development, and government funding. The factor with the greatest contribution to UIR was the level of economic development, with the levels of urbanization and infrastructure investment gradually changing from exerting positive spillover effects to exerting negative spillover effects.

### Suggestions

Based on the above findings, this paper makes the following suggestions. First, economic development and government fiscal investment have significant positive spatial spillover impacts on UIR levels. Economic development promotes the diversification of industrial structure and drives innovation capacity, further increasing the resistance, resilience and adjustment of cities to various types of crises, while promoting the level of infrastructure development and resilience. The size of government finances and the appropriateness of the expenditure structure directly affect the concentration of social resource allocation, which in turn has an impact on the resilience of urban infrastructure. Therefore, expanding the scale of economic development and increasing the share of government financial expenditure on infrastructure construction will further expand the scale of infrastructure construction and strive to improve the overall level and quality of infrastructure construction, thereby increasing the level of infrastructure resilience. On the one hand, we should broaden the investment and financing channels for infrastructure construction in central, western and northeast-ern China, and continuously improve the capital market, so as to expand the investment in infrastructure and its construction scale; on the other hand, we should reduce the duplication of infrastructure construction in the eastern region, strengthen the maintenance of existing facilities and improve their utilization efficiency. We should enhance the overall level of infrastructural resilience in cities and reduce regional disparities to further promote balanced urban development.

Second, industrial structures have a significant positive spatial spillover effect on the level of UIR. As the driving force of economic development, a favorable industrial structure can bring strong development momentum to economic development and lay a solid foundation for infrastructure construction. Optimise the industrial structure, accelerate the development of modern service industries, improve the efficiency and quality of service industries, and accelerate the scale of emerging service industries in central, western, and northeastern China, with an emphasis on the development of services such as recreation and health, information and

consulting, information technology services, exhibitions and the cultural industry. Vigorously promote the development of high-tech industries, cultivate new dynamism, achieve the conversion of old and new dynamism, promote high-quality economic development, and improve the level of UIR.

Third, urbanization has a significant negative spatial spillover effect on the level of UIR. Urbanization brings about a high concentration of social capital and human resources, providing social and human resources for urban infrastructure construction, while rapid urbanization also puts enormous pressure on infrastructure, with uncontrolled population growth greatly exceeding the carrying capacity of urban infrastructure. Therefore, we should reasonably control the direction and scale of urbanization development, enhance the endogenous development momentum of the whole UIR by means of a reasonable layout of population density, etc., give full play to the positive effect of public resource allocation on infrastructure construction and the spillover effect of technology, break down barriers to the flow of production factors and technology, and take other measures to avoid the siphoning effect brought about by urbanization.

## Discussion

### Comparison with previous studies

Previous studies have shown that there are complex spatial and temporal dynamics underlying the resilience of urban infrastructure, as evidenced by the exploration of its spatial and temporal divergence [6, 18–27, 29–35]. However, was a spatial spillover effect of UIR observed? The main objective of this study was to identify the factors influencing UIR and further analyze whether spatial spillover effects are produced from these factors. Our research results revealed that the degree of the spatial spillover effect of each influencing factor on UIR differed, and both positive and negative spillover effects were observed. Specifically, the spillover effects were mainly influenced by factors such as the population concentration, financial and economic development, and government funding. The factor with the greatest contribution to UIR is the level of economic development, and the level of urbanization and infrastructure investment gradually changed from positive to negative spillover effects.

Despite the few studies published in this field, some useful explorations have been conducted. Using Northwest China as the study area, Gao et al. confirmed that finance, market consumption level, and industrial structure exert significant and positive directional effects on UIR [23]. Zhang et al. showed that government funding exerts a significant positive effect on UIR [18]. These results were consistent with the results from our study. Existing studies have mainly focused on exploring the factors influencing the resilience of urban infrastructure in a particular region. However, previous studies have not revealed the spatial spillover effects of UIR, particularly an analysis comparing the spatial spillover effects between different years at the urban scale in China. Therefore, the impact of spatial spillover effects on UIR deserves further study.

### Limitations

Clarification of the characteristics of the dynamic evolution of the UIR and the spillover effects is important to enhance urban disaster resilience and improve sustainable urban development. In national urban infrastructure construction, the overall level of the infrastructure resilience of cities should be improved, and the gap should be narrowed to promote the balanced development of cities. Economic development drives the rapid development of urban and rural infrastructure, and research on infrastructure resilience at different scales might better reflect its spatial heterogeneity and the degree of integrated development. The research unit of this

paper is national prefecture-level cities and above, and the research scale is relatively macro. Therefore, future research can build a long-term optimization path for different scales. Urban blue-green infrastructure is an important part of urban infrastructure [58–60], which includes water systems, greenways, wetlands, and forests. It is a crucial aspect affecting the thermal environment of the city, influencing urban spatial layout, and increasing urban resilience [61, 62]. Although the index system we constructed includes some blue–green infrastructure because cities at the prefecture level and above include different provinces and regions with different natural, economic, and social development conditions and obvious development differentiation, data on certain blue–green infrastructure indicators are difficult to obtain for most cities at the municipal scale. Therefore, future research should address the crucial role of blue–green infrastructure in UIR based on the premise of data availability, and external impacts tailored to local development circumstances should be chosen for different locations to acquire more generalizable insights.

## Supporting information

**S1 Table. Raw data for measuring the resilience level of urban infrastructure.** This includes raw data on 25 urban infrastructure resilience evaluation indicators for 283 prefecture-level and above cities in China in 2010, 2015 and 2019.
(XLSX)

**S2 Table. Raw data on the explanatory variables of the spatial spillover effect of urban infrastructure resilience.** This includes raw data on eight explanatory variables for the spatial spillover effects of urban infrastructure resilience for 283 prefecture-level and above cities in China in 2010, 2015, and 2019.
(XLSX)

## Acknowledgments

We greatly appreciated Prof. Jingfeng Ge and Prof. Zhongjiang Feng for their valuable comments at various stages of this study. We thank AJE (www.aje.cn) for its linguistic assistance during the preparation of this manuscript.

## Author Contributions

**Conceptualization:** Hao Wang, Zhiying Huang, Yanqing Liang, Xiangyun An.

**Data curation:** Hao Wang, Zhiying Huang, Yanqing Liang, Qingxi Zhang.

**Formal analysis:** Hao Wang, Zhiying Huang, Yanqing Liang.

**Funding acquisition:** Yanqing Liang.

**Methodology:** Hao Wang, Zhiying Huang, Xiangyun An.

**Project administration:** Yanqing Liang.

**Resources:** Yanqing Liang.

**Software:** Hao Wang, Yanqing Liang.

**Writing – original draft:** Hao Wang.

**Writing – review & editing:** Hao Wang, Zhiying Huang, Yanqing Liang, Shaoxiong Hu, Liye Cui.

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
