## [Decision Letter · Decision Letter 0]

28 Nov 2022

PONE-D-22-30335Dynamic evolution of urban infrastructure resilience and its spatial spillover effects: An empirical study from ChinaPLOS ONE

Dear Dr. Liang,

Thank you for submitting your manuscript to PLOS ONE. After careful consideration, we feel that it has merit but does not fully meet PLOS ONE’s publication criteria as it currently stands. Therefore, we invite you to submit a revised version of the manuscript that addresses the points raised during the review process.

We look forward to receiving your revised manuscript.

Kind regards,

Jun Yang

Academic Editor

PLOS ONE

Journal Requirements:

"This research was funded by National Natural Science Foundation of China (41471090), Yanqing Liang. Key Projects of Science and Technology Research of Hebei Provincial Education Department (ZD2019115), Yanqing Liang."

3. We note that Figures 3 and 5 in your submission contain map images which may be copyrighted. All PLOS content is published under the Creative Commons Attribution License (CC BY 4.0), which means that the manuscript, images, and Supporting Information files will be freely available online, and any third party is permitted to access, download, copy, distribute, and use these materials in any way, even commercially, with proper attribution. For these reasons, we cannot publish previously copyrighted maps or satellite images created using proprietary data, such as Google software (Google Maps, Street View, and Earth). For more information, see our copyright guidelines: http://journals.plos.org/plosone/s/licenses-and-copyright.

a. You may seek permission from the original copyright holder of Figures 3 and 5 to publish the content specifically under the CC BY 4.0 license.  

Additional Editor Comments:

Reviewer 1

The authors investigated dynamic evolution of urban infrastructure resilience and its spatial spillover effects in China. The research methodologies are reasonable, and the findings are interesting. However, there are still some aspects that should be improved to make the paper publishable. I focus here only on some points, which are hopefully easy for the authors to take into account in the revision.

(1) Line 21-Kernel density estimation? (2) Part Introduction - The innovation should be highlighted. There are some references on this topic, I suggest you supplied it in this part, as follows. 1) Contribution of urban functional zones to the spatial distribution of urban thermal environment, Building and Environment (2022), doi: https://doi.org/10.1016/j.buildenv.2022.109000.

2) Exploring thermal comfort of urban buildings based on local climate zones, Journal of Cleaner Production (2022), doi: https://doi.org/10.1016/j.jclepro.2022.130744.

3) Urban scale ventilation analysis based on neighborhood normalized current model, Sustainable Cities and Society (2022), doi: https://doi.org/10.1016/j.scs.2022.103746

4) Contribution of urban ventilation to the thermal environment and urban energy demand: Different climate background perspectives, Science of the Total Environment (2021), https://doi.org/10.1016/j.scitotenv.2021.148791. 5) Suitability of human settlements in mountainous areas from the perspective of ventilation: a case study of the main urban area of Chongqing, Journal of Cleaner Production(2021), https://doi.org/10.1016/j.jclepro.2021.127467.

6) How to promote the transition from solo driving to mobility services delivery? An empirical study focusing on ridesharing, Transport Policy, 2022(10), https://doi.org/10.1016/j.tranpol.2022.10.009.

7) The impact of urban renewal on land surface temperature changes: A case study in the main city of Guangzhou, China. Remote Sensing (2020), https://doi.org/10.3390/rs12050794.

8) Impacts of Neighboring Buildings on the Cold Island Effect of Central Parks: A Case Study of Beijing, China. Sustainability (2020), doi: https://doi.org/10.3390/su12229499.

9) Impacts of Urban Green Space on Land Surface Temperature from Urban Block Perspectives. Remote Sensing (2022), doi: https://doi.org/10.3390/rs14184580.

(3) Sec Construction of the Urban Infrastructure Resilience Index System - 25 indicators were used in this study, please explain the reason why you used it. Importantly, how to ensure the accuracy of the results? (4) Sec Conclusions - Summarize the main conclusions. (5) Discussion is lacking, maybe comparison and limitation should be discussed.

This manuscript presents an interesting topic. The following issues still need to be further improved and explained:

1. This is an interesting study based on extensive data analysis materials. The authors should focus on improving the readability of the paper, in particular, we should sort out Introduction section on the inner and outer limits of urban infrastructure.

2. Another aspect that the authors should improve on is to make the paper with more international and theoretical relevance, the main issue is the theoretical logic of resilience theory and urban infrastructure

3. As a specification manuscript, there are some shortcomings, there are some specific problems in the manuscript, such as the nine segment line display of China in Figure 3 and the failure to mark the Diaoyu Islands.

4. What is the reason for the selection of the manuscripts for every decade after 2000? Policy context or particularly profound reasons in manuscripts.

5. Is the choice of indicator system scientific? Why is blue-green infrastructure not considered as an important element of urban resilience?

6. It is recommended to add some references, for example, “Spatial-Temporal Patterns of Network Structure of Human Settlements Competitiveness in Resource-Based Urban Agglomerations”，“Spatial Responses of Ecosystem Service Value during the Development of Urban Agglomerations ” and “Morphological and functional polycentric structure assessment of megacity: An integrated approach with spatial distribution and interaction”.

This manuscript presents an interesting topic. The following issues still need to be further improved and explained:

1. This is an interesting study based on extensive data analysis materials. The authors should focus on improving the readability of the paper, in particular, we should sort out Introduction section on the inner and outer limits of urban infrastructure.

2. Another aspect that the authors should improve on is to make the paper with more international and theoretical relevance, the main issue is the theoretical logic of resilience theory and urban infrastructure

3. As a specification manuscript, there are some shortcomings, there are some specific problems in the manuscript, such as the nine segment line display of China in Figure 3 and the failure to mark the Diaoyu Islands.

4. What is the reason for the selection of the manuscripts for every decade after 2000? Policy context or particularly profound reasons in manuscripts.

5. Is the choice of indicator system scientific? Why is blue-green infrastructure not considered as an important element of urban resilience?

6. It is recommended to add some references, for example, “Spatial-Temporal Patterns of Network Structure of Human Settlements Competitiveness in Resource-Based Urban Agglomerations”，“Spatial Responses of Ecosystem Service Value during the Development of Urban Agglomerations ” and “Morphological and functional polycentric structure assessment of megacity: An integrated approach with spatial distribution and interaction”.

Reviewers' comments:

Reviewer's Responses to Questions

**Comments to the Author**

1. Is the manuscript technically sound, and do the data support the conclusions?

Reviewer #1: Partly

Reviewer #2: Partly

2. Has the statistical analysis been performed appropriately and rigorously? 

Reviewer #1: Yes

Reviewer #2: Yes

3. Have the authors made all data underlying the findings in their manuscript fully available?

Reviewer #1: Yes

Reviewer #2: Yes

4. Is the manuscript presented in an intelligible fashion and written in standard English?

Reviewer #1: Yes

Reviewer #2: Yes

5. Review Comments to the Author

Reviewer #1: The authors investigated dynamic evolution of urban infrastructure resilience and its spatial spillover effects in China. The research methodologies are reasonable, and the findings are interesting. However, there are still some aspects that should be improved to make the paper publishable. I focus here only on some points, which are hopefully easy for the authors to take into account in the revision.

(1) Line 21-Kernel density estimation?

(2) Part Introduction - The innovation should be highlighted. There are some references on this topic, I suggest you supplied it in this part, as follows.

1)Contribution of urban functional zones to the spatial distribution of urban thermal environment, Building and Environment (2022), doi: https://doi.org/10.1016/j.buildenv.2022.109000.

2)Exploring thermal comfort of urban buildings based on local climate zones, Journal of Cleaner Production (2022), doi: https://doi.org/10.1016/j.jclepro.2022.130744.

3)Urban scale ventilation analysis based on neighborhood normalized current model, Sustainable Cities and Society (2022), doi: https://doi.org/10.1016/j.scs.2022.103746

4)Contribution of urban ventilation to the thermal environment and urban energy demand: Different climate background perspectives, Science of the Total Environment (2021), https://doi.org/10.1016/j.scitotenv.2021.148791.

5)Suitability of human settlements in mountainous areas from the perspective of ventilation: a case study of the main urban area of Chongqing, Journal of Cleaner Production(2021), https://doi.org/10.1016/j.jclepro.2021.127467.

6)How to promote the transition from solo driving to mobility services delivery? An empirical study focusing on ridesharing, Transport Policy, 2022(10), https://doi.org/10.1016/j.tranpol.2022.10.009.

7) The impact of urban renewal on land surface temperature changes: A case study in the main city of Guangzhou, China. Remote Sensing (2020), https://doi.org/10.3390/rs12050794.

8)Impacts of Neighboring Buildings on the Cold Island Effect of Central Parks: A Case Study of Beijing, China. Sustainability (2020), doi: https://doi.org/10.3390/su12229499.

9)Impacts of Urban Green Space on Land Surface Temperature from Urban Block Perspectives. Remote Sensing (2022), doi: https://doi.org/10.3390/rs14184580.

(3) Sec Construction of the Urban Infrastructure Resilience Index System - 25 indicators were used in this study, please explain the reason why you used it. Importantly, how to ensure the accuracy of the results?

(4) Sec Conclusions - Summarize the main conclusions.

(5) Discussion is lacking, maybe comparison and limitation should be discussed.

Reviewer #2: This manuscript presents an interesting topic. The following issues still need to be further improved and explained:

1.This is an interesting study based on extensive data analysis materials. The authors should focus on improving the readability of the paper, in particular, we should sort out Introduction section on the inner and outer limits of urban infrastructure.

2.Another aspect that the authors should improve on is to make the paper with more international and theoretical relevance, the main issue is the theoretical logic of resilience theory and urban infrastructure

3.As a specification manuscript, there are some shortcomings, there are some specific problems in the manuscript, such as the nine segment line display of China in Figure 3 and the failure to mark the Diaoyu Islands.

4.What is the reason for the selection of the manuscripts for every decade after 2000? Policy context or particularly profound reasons in manuscripts.

5. Is the choice of indicator system scientific? Why is blue-green infrastructure not considered as an important element of urban resilience?

6. It is recommended to add some references, for example, “Spatial-Temporal Patterns of Network Structure of Human Settlements Competitiveness in Resource-Based Urban Agglomerations”，“Spatial Responses of Ecosystem Service Value during the Development of Urban Agglomerations ” and “Morphological and functional polycentric structure assessment of megacity: An integrated approach with spatial distribution and interaction”.

6. PLOS authors have the option to publish the peer review history of their article (what does this mean?). If published, this will include your full peer review and any attached files.

Reviewer #1: No

Reviewer #2: No

---

## [Author Response · Author response to Decision Letter 0]

5 Jan 2023

Reply to-comments

Manuscript Number: PONE-D-22-30335

Summary of the major changes and responses to the Reviewers’ comments: 

Dear Editor,

On behalf of my co-authors, we thank you very much for giving us an opportunity to revise our manuscript again. We greatly appreciate the editors for their positive constructive comments and suggestions on our manuscript entitled “Dynamic evolution of urban infrastructure resilience and its spatial spillover effects: An empirical study from China”. (ID: PONE-D-22-30335).

We have studied the editors’ comments carefully and addressed them one by one. Attached please find the response to the editor and revised manuscript, which we would like to submit for your kind consideration. All modifications in the manuscript are indicated in red font.

As requested by your journal, we have added a statement of the funder's role to the fourth paragraph of the cover letter. The administrative maps of China in Figures 3 and Figures 5 of this study all are audited with the review number GS (2019)1822, and both are open access. There is no copyright protection problem.

We would like to express our great appreciation to you for your comments on our paper. We look forward to hearing from you.

Thank you and best regards.

Yours sincerely,

Hao Wang

School of Geographic Sciences, Hebei Normal University, Shijiazhuang 050024, China

Tel: 86-031180787632; E-mail: liangyanqing@126.com

 

Responses to Anonymous Reviewer #1’s Comments:

Specific comments:

The authors investigated dynamic evolution of urban infrastructure resilience and its spatial spillover effects in China. The research methodologies are reasonable, and the findings are interesting. However, there are still some aspects that should be improved to make the paper publishable. I focus here only on some points, which are hopefully easy for the authors to take into account in the revision.

Response: On behalf of my co-authors, we greatly appreciate your positive and constructive comments and suggestions on our manuscript. We have carefully studied the relevant comments and revised the manuscript according to each comment, particularly the presentation. We revised the manuscript and visualized the relevant results according to your concerns. All modifications are indicated in red font. The point-by-point responses are presented below.

1. Line 21-Kernel density estimation?

Response: Thank you for your detailed suggestions. We apologize for the inadvertent error and have changed "Kernel density estimation" to "kernel density estimation" and verified similar issues throughout the text.

2. Part Introduction - The innovation should be highlighted. There are some references on this topic, I suggest you supplied it in this part, as follows. 

Response: Thank you for your pertinent suggestions. In accordance with your suggestions, we have further emphasized the innovative points of the paper, which mainly focus on three aspects: the construction of the indicator system, research perspective, and research scale. We also greatly appreciate the recommendation of related literature, which we have fully referenced and incorporated in the Introduction section. The specific information is as follows (Page 3-4, lines 117-132):

“The innovations are as follows. (1) Construction of a new evaluation index for UIR: Fewer studies have been published that examine infrastructure resilience as a research topic, and the indicator system is not sufficiently considered in terms of environmental facilities and cultural and health facilities. Based on the dissection of urban infrastructure and its resilience connotation, this paper evaluates the resilience of urban infrastructure from six perspectives to make the evaluation system more comprehensive and the evaluation results more accurate: resilience of energy facilities, resilience of supply and drainage facilities, resilience of transportation facilities, resilience of postal and communication facilities, resilience of environmental facilities and resilience of cultural and health facilities. (2) Research perspective: This paper uses a spatial econometric model to explore the spatial spillover effects of UIR. Compared with the traditional linear regression model, the spatial econometric model is a spatial model that is closer to the actual situation and includes the degree of the spatial effect of each variable. (3) Research scale: Most of the existing studies analyze the UIR of a particular region; this paper selects 283 cities at the prefecture level and above in China. A larger sample might highlight the advantages of assessing the UIR and produce more accurate results, facilitating differentiated cross-sectional comparisons between cities within different administrative regions.”

3. Sec Construction of the Urban Infrastructure Resilience Index System - 25 indicators were used in this study, please explain the reason why you used it. Importantly, how to ensure the accuracy of the results? 

Response: Thank you for your suggestions. Based on your suggestions, the reasons for the selection of indicators are clarified in the Construction of the Urban Infrastructure Resilience Index System section of the paper. The details are as follows (page 4-5, lines 157-172):

(1) Regarding the reasons for using 25 indicators in this study

The 25 indicators chosen in this study are mainly considered from the perspectives of whether the indicators fully reflect the connotation of urban infrastructure resilience, whether they comprehensively consider the relevant indicators covered by the existing relevant literature, and the availability of data. The specific reasons are as follows.

① Considerations from the connotation of infrastructure resilience

Infrastructure resilience emphasizes its state and ability to resist external perturbations. The 25 indicators used in this study, such as the gas penetration rate, total electricity consumption, water supply pipeline density, and drainage pipeline density, were selected because they are closely related to the connotation of infrastructure resilience.

② Consideration of infrastructure resilience indicators based on references to the relevant literature

The current index system for urban infrastructure and its resilience is mainly constructed using energy, water supply and drainage, transportation, communication and other municipal utilities. For example, Chen et al. described the state of infrastructure resilience in terms of the gas penetration rate, water penetration rate, and road area per capita[6]. Liu, Chen, Tao, Bai, and Zhang et al. constructed indicator systems including daily domestic water consumption per capita, water supply pipeline density, drainage pipeline density, and road area per capita[5,7-13,26,27,40-42].

Using the principles of comprehensiveness, importance, measurability, and availability of evaluation indicators, we used the “Standard for Basic Terminology of Urban Planning” (GB/T50280-98) as the basis to count the frequency of indicators involved in 32 highly relevant papers, among which 21 indicators have a high usage rate (as shown in Table 1), and all these high-frequency indicators were included in the 25 indicators used in this study. Thus, we have selected four indicators, namely, special vehicles for sanitation for 10000 people, the number of secondary schools, the number of primary schools, and library collections per 100 people, as useful supplements to the engineering facilities and social service facilities in the connotation of infrastructure together to ensure that the index system is more comprehensive.

③ Considerations based on data availability

The research object of this study is 283 cities at the prefecture level and above in China, the sample size is large, and the data on indicators such as per capita shelter space area and emergency personnel ratio are not available. Therefore, after considering the comprehensiveness and availability of data, 25 representative indicators with a high utilization rate were identified to construct the urban infrastructure resilience evaluation system.

Table 1. Comprehensive evaluation index system of urban infrastructure resilience.

Index Property Frequency Weight

Gas penetration rate Positive 32 0.043

Total gas supply Positive 28 0.026

Total electricity consumption Positive 25 0.040

Water supply pipeline density Positive 32 0.057

Water penetration rate Positive 28 0.037

Daily domestic water consumption per capita Negative 30 0.043

Drainage pipeline density Positive 32 0.047

Road area per capita Positive 32 0.048

Road network density Positive 29 0.047

Number of buses per 10,000 people Positive 31 0.032

Total number of bus passengers Positive 28 0.039

Number of households with internet access per 10,000 people Positive 30 0.061

Number of households with cell phones per 10,000 people Positive 32 0.033

Telecommunications services per capita Positive 29 0.035

Postal business per capita Positive 28 0.036

Centralized urban sewage treatment rate Positive 25 0.036

Harmless disposal rate of domestic waste Positive 26 0.020

Special vehicles for sanitation for 10,000 people Positive 10 0.030

Green space per capita Positive 29 0.047

Greening coverage of built-up areas Positive 25 0.044

Number of secondary schools Positive 5 0.040

Number of primary schools Positive 4 0.049

Library collections per 100 people Positive 10 0.042

Number of hospital beds per 1,000 population Positive 26 0.036

Number of health institutions Positive 29 0.032

(2) Considerations of the accuracy of the results from the indicator system

Based on the aforementioned analysis of the reasons for using the 25 indicators described in this study, the indicators selected in this study are mainly related to the connotation of urban infrastructure resilience, covering the high-frequency indicators involved in the indicator systems used in related studies and considering the availability of data and other parameters, which has met the need for accurate results to the greatest extent. We believe the results are accurate.

“The current index system for urban infrastructure and its resilience is mainly constructed in terms of energy, water supply and drainage, transportation, communication and other municipal utilities. For example, Chen et al. described the state of infrastructure resilience in terms of the gas penetration rate, water penetration rate, and road area per capita [6]. Liu, Chen, Tao, Bai, and Zhang et al. constructed indicator systems including the daily domestic water consumption per capita, water supply pipeline density, drainage pipeline density, and road area per capita [5,7-13,26,27,40-42].

This study follows the principles of comprehensiveness, importance, measurability, and availability of evaluation indicators and uses the “Standard for Basic Terminology of Urban Planning” (GB/T50280-98) as the basis for selection. The frequency of the indicators involved in 32 highly relevant papers is counted, and 21 indicators with representative and high usage are selected. Thus, we have selected four indicators, namely, special vehicles for sanitation for 10000 people, the number of secondary schools, the number of elementary schools, and library collections per 100 people, as useful supplements to the engineering facilities and social service facilities in the connotation of infrastructure together to ensure that the index system is more comprehensive. The finalized urban infrastructure resilience evaluation index system consists of 25 indicators such as the gas penetration rate, total gas supply, total electricity consumption, and water supply pipeline density (Table 1).”

Table 1. Comprehensive evaluation index system of urban infrastructure resilience.

Sub-index Index Property Frequency Weight

Resilience of energy facilities Gas penetration rate Positive 32 0.043

 Total gas supply Positive 28 0.026

 Total electricity consumption Positive 25 0.040

Resilience of supply and drainage facilities Water supply pipeline density Positive 32 0.057

 Water penetration rate Positive 28 0.037

 Daily domestic water consumption per capita Negative 30 0.043

 Drainage pipeline density Positive 32 0.047

Resilience of transport facilities Road area per capita Positive 32 0.048

 Road network density Positive 29 0.047

 Number of buses per 10,000 people Positive 31 0.032

 Total number of bus passengers Positive 28 0.039

Resilience of postal and communication facilities Number of households with internet access per 10,000 people Positive 30 0.061

 Number of households with cell phones per 10,000 people Positive 32 0.033

 Telecommunications services per capita Positive 29 0.035

 Postal business per capita Positive 28 0.036

Resilience of environmental facilities Centralized urban sewage treatment rate Positive 25 0.036

 Harmless disposal rate of domestic waste Positive 26 0.020

 Special vehicles for sanitation for 10,000 people Positive 10 0.030

 Green space per capita Positive 29 0.047

 Greening coverage of built-up areas Positive 25 0.044

Resilience of cultural and health facilities Number of secondary schools Positive 5 0.040

 Number of primary schools Positive 4 0.049

 Library collections per 100 people Positive 10 0.042

 Number of hospital beds per 1,000 population Positive 26 0.036

 Number of health institutions Positive 29 0.032

The references are as follows:

[5] Liu QQ, Wang SJ, Zhang WZ, Li JM, Zhao YB, Li W. China's municipal public infrastructure: Estimating construction levels and investment efficiency using the entropy method and a DEA model. Habitat International. 2017; 64: 59-70. https://doi.org/10.1016/j.habitatint.2017.04.010

[6] Chen M, Jiang Y, Wang ED, Wang Y, Zhang J. Measuring Urban Infrastructure Resilience via Pressure-State-Response Framework in Four Chinese Municipalities. Applied Sciences. 2022; 12:2819. https://doi.org/10.3390/app12062819

[7] Heshmati A, Rashidghalam M. Measurement and analysis of urban infrastructure and its effects on urbanization in China. Journal of Infrastructure Systems. 2020; 26: 04019030. https://doi.org/10.1061/(ASCE)IS.1943-555X.0000513 

[8] Chen HL, Zhang Y, Zhang NX, Zhou M, Ding HP. Analysis on the Spatial Effect of Infrastructure Development on the Real Estate Price in the Yangtze River Delta. Sustainability. 2022; 14:7569. https://doi.org/10.3390/su14137569

[9] Tao ZM. Research on the degree of coupling between the urban public infrastructure system and the urban economic, social, and environmental system: A case study in Beijing, China. Mathematical Problems in Engineering. 2019; https://doi.org/10.1155/2019/8206902

[10] Lu C, Hong WX, Wang YT, Zhao DF, Study on the Coupling Coordination of Urban Infrastructure and Population in the Perspective of Urban Integration. IEEE Access. 2021; 9: 124070-124086, https://doi.org/10.1109/ACCESS.2021.3110368

[11] Yu HS, Yang J, Li T, Jin Y, Sun DQ. Morphological and functional polycentric structure assessment of megacity: An integrated approach with spatial distribution and interaction. Sustainable Cities and Society. 2022; 80: 103800. https://doi.org/10.1016/j.scs.2022.103800

[12] Yu WB, Yang J, Sun DQ, Yu HS, Yao Y, Xiao XM, et al. Spatial-temporal patterns of network structure of human settlements competitiveness in resource-based urban agglomerations. Frontiers in Environmental Science. 2022; 647. https://doi.org/10.3389/fenvs.2022.893876

[13] Zhang X, Zhong SQ, Ling S, Jia N, Qi H, He ZB. How to promote the transition from solo driving to mobility services delivery? An empirical study focusing on ridesharing. Transport Policy. 2022; 129: 176-187. https://doi.org/10.1016/j.tranpol.2022.10.009

……

4. Sec Conclusions - Summarize the main conclusions. 

Response: We are very grateful for your suggestions and agree with your comments. We summarize the main findings of the study results with respect to the spatio-temporal dynamic evolution characteristics and spatial spillover effects. The specific information is as follows (Page 13, lines 459-474):

“This paper explored the dynamic evolution of the spatial and temporal characteristics of UIR using kernel density estimation and ESDA methods for 283 prefecture-level and above cities in China. The optimal spatial econometric model was selected to explore its spatial spillover effects.

Our results showed that the level of UIR in China tended to increase over the study period. The eastern region had the highest UIR. Regions above the high resilience level showed a tendency to spread from coastal to inland areas, and regions below the moderate resilience level showed a degree of agglomeration. Cities with similar levels of infrastructure resilience exhibited a more concentrated spatial distribution, and the spatial distribution pattern from coastal to inland areas was a hotspot-subhotspot-subcoldspot-coldspot pattern.

In terms of the spatial spillover effects of UIR, different spatial dependencies on infrastructure resilience among cities were observed, and the spillover effects were prominent. Various factors had different degrees of influence on the spillover effects of UIR, and the spillover effects were mainly influenced by the population concentration, financial and economic development, and government funding. The factor with the greatest contribution to UIR was the level of economic development, with the levels of urbanization and infrastructure investment gradually changing from exerting positive spillover effects to exerting negative spillover effects.”

5. Discussion is lacking, maybe comparison and limitation should be discussed.

Response: Thank you for your detailed suggestions. Based on your suggestions, We have added the Discussion section as a new chapter to make the paper more complete and logical. In the Discussion section, the main results, such as the spatial spillover effects of urban infrastructure resilience and the extent of their impacts, are compared with the main ideas presented in the related literature, and the limitations of the research scale and evaluation index system are discussed. The details are as follows (Pages 14-15, lines 513-554):

Discussion

Comparison with Previous Studies

“Previous studies have shown that there are complex spatial and temporal dynamics underlying the resilience of urban infrastructure, as evidenced by the exploration of its spatial and temporal divergence [6,18-27,29-35]. However, was a spatial spillover effect of UIR observed? The main objective of this study was to identify the factors influencing UIR and further analyze whether spatial spillover effects are produced from these factors. Our research results revealed that the degree of the spatial spillover effect of each influencing factor on UIR differed, and both positive and negative spillover effects were observed. Specifically, the spillover effects were mainly influenced by factors such as the population concentration, financial and economic development, and government funding. The factor with the greatest contribution to UIR is the level of economic development, and the level of urbanization and infrastructure investment gradually changed from positive to negative spillover effects.

Despite the few studies published in this field, some useful explorations have been conducted. Using Northwest China as the study area, Gao et al. confirmed that finance, market consumption level, and industrial structure exert significant and positive directional effects on UIR [23]. Zhang et al. showed that government funding exerts a significant positive effect on UIR[18]. These results were consistent with the results from our study. Existing studies have mainly focused on exploring the factors influencing the resilience of urban infrastructure in a particular region. However, previous studies have not revealed the spatial spillover effects of UIR, particularly an analysis comparing the spatial spillover effects between different years at the urban scale in China. Therefore, the impact of spatial spillover effects on UIR deserves further study.

Limitations

Clarification of the characteristics of the dynamic evolution of the UIR and the spillover effects is important to enhance urban disaster resilience and improve sustainable urban development. In national urban infrastructure construction, the overall level of the infrastructure resilience of cities should be improved, and the gap should be narrowed to promote the balanced development of cities. Economic development drives the rapid development of urban and rural infrastructure, and research on infrastructure resilience at different scales might better reflect its spatial heterogeneity and the degree of integrated development. The research unit of this paper is national prefecture-level cities and above, and the research scale is relatively macro. Therefore, future research can build a long-term optimization path for different scales. Urban blue-green infrastructure is an important part of urban infrastructure [58-60], which includes water systems, greenways, wetlands, and forests. It is a crucial aspect affecting the thermal environment of the city, influencing urban spatial layout, and increasing urban resilience [61,62]. Although the index system we constructed includes some blue‒green infrastructure because cities at the prefecture level and above include different provinces and regions with different natural, economic, and social development conditions and obvious development differentiation, data on certain blue‒green infrastructure indicators are difficult to obtain for most cities at the municipal scale. Therefore, future research should address the crucial role of blue‒green infrastructure in UIR based on the premise of data availability, and external impacts tailored to local development circumstances should be chosen for different locations to acquire more generalizable insights.”

The references are as follows:

[18] Zhang P, Yu W, Zhang YW. Spatial-temporal differentiation and its influencing factors of Shandong province's urban resilience. Urban Problems. 2018; 9: 27-34.

[19] Zhu JQ, Sun HX. Research on spatial-temporal evolution and influencing factors of urban resilience of China's three metropolitan agglomerations. Soft Science. 2020; 34: 72-79.

[20] Zhang YH, Xue Y, Xu ML, Li F. Research on the dynamic prediction and spatial-temporal evolution of urban resilience. Modernization of Management. 2021; 41: 77-81. 

[21] Chen XH, Lou JN, Wang Y. Research on the evolution of spatial and temporal patterns and dynamic simulation of urban resilience in the Harbin-Changzhou urban agglomeration. Scientia Geographica Sinica. 2020; 40: 2000-2009. 

[22] Zhang MD, Li WL. Spatial difference and convergence of urban resilience level in northeast China. Journal of Industrial Technological Economics. 2020; 39: 3-12.

[23] Gao ZG, Ding YY. Measurement of urban resilience and its influencing factors in northwest China. Science & Technology Review. 2021; 39: 118-129.

[24] Zhou Q, Liu DL. Study on the coordinated development of urban resilience and urbanization level in the urban agglomeration of Yangtze River delta. Research of Soil and Water Conservation. 2020; 27: 286-292. 

[25] Bai S, Wu JH, Wang Z. Coupling relationship between urban resilience and land use efficiency in He' nan province. Bulletin of Soil and Water Conservation. 2022; 42: 308-316.

[26] Bai LM, Xiu CL, Feng XH, Mei DW, Wei Y. A comprehensive assessment of urban resilience and its spatial differentiation in China. World Regional Studies. 2019; 28: 77-87.

[27] Zhang MD, Feng XQ. Comprehensive evaluation on Chinese cities' resilience. Urban Problems. 2018; 10: 27-36.

……

[61] Ren JY, Yang J, Zhang YQ, Xiao XM, Xia JH, Li XM, et al. Exploring thermal comfort of urban buildings based on local climate zones. Journal of Cleaner Production. 2022; 340:130744. https://doi.org/10.1016/j.jclepro.2022.130744

[62] An HM, Cai HY, Xu XL, Qiao Z, Han DR. Impacts of Urban Green Space on Land Surface Temperature from Urban Block Perspectives. Remote Sensing. 2022; 14:4580. https://doi.org/10.3390/rs14184580

Once again, thank you very much for your constructive comments and suggestions, which have helped us a lot to improve the quality of the paper.

 

Responses to Anonymous Reviewer #2’s Comments:

Specific comments:

This manuscript presents an interesting topic. The following issues still need to be further improved and explained:

Response: On behalf of my co-authors, we greatly appreciate your positive constructive comments and suggestions on our manuscript. We have carefully studied the relevant comments and revised the manuscript according to each comment, particularly the presentation. We revised the manuscript and visualized the relevant results based on your suggestions. All modifications are indicated in red font. The point-by-point responses are presented below.

1. This is an interesting study based on extensive data analysis materials. The authors should focus on improving the readability of the paper, in particular, we should sort out Introduction section on the inner and outer limits of urban infrastructure. 

Response: Thank you for your suggestions. According to your suggestions, we have carefully sorted the studies related to the composition and classification of urban infrastructure and found that the delineation of the boundaries of urban infrastructure is a complex scientific issue. The existing studies do not clarify how their inner and outer boundaries are divided; only transportation facilities are divided into inner urban transportation facilities and outer urban transportation facilities. The research object of this study is 283 prefecture-level and above cities in China, and we used the data for each indicator from the “National Statistical Yearbook”, whose statistical caliber is urban, which means the default scope is within the urban area, but the inner and outer boundaries of urban infrastructure are not clearly defined in the statistical yearbook. Thus, as statistics of the corresponding caliber meet the needs of this study, we only define the connotation of the inner and outer boundaries of urban infrastructure as the use of infrastructure and services outside the boundaries of a particular urban administrative district. The details are as follows (Page 2, lines 54-63):

“Studies assessing the composition and classification of urban infrastructure have all classified it into six major systems and their combinations [3-13]: water supply and drainage, energy supply, road transport, postal and telecommunications, landscaping and sanitation, and disaster prevention facilities. However, these studies have not clearly described internal and external boundaries, and only the transport facilities system is divided between inner and outer urban transport facilities. We consider the internal and external boundaries of urban infrastructure as the use of infrastructure and services outside the administrative boundaries of the city. Thus, we define the research object of this paper as the infrastructure of energy, supply and drainage, transportation, postal, and telecommunications within the city, with the boundary of administrative districts.”

The references are as follows:

[3] Guo GQ. Construction and development of urban infrastructure in China. 1st ed.; China Architecture & Building Press: Beijing, China, 1990; pp. 3-4

[4] Xie WH, Deng W. Urban Economics. 1st ed.; Tsinghua University Press: Beijing, China, 1996; pp. 309-310

[5] Liu QQ, Wang SJ, Zhang WZ, Li JM, Zhao YB, Li W. China's municipal public infrastructure: Estimating construction levels and investment efficiency using the entropy method and a DEA model. Habitat International. 2017; 64: 59-70. https://doi.org/10.1016/j.habitatint.2017.04.010

[6] Chen M, Jiang Y, Wang ED, Wang Y, Zhang J. Measuring Urban Infrastructure Resilience via Pressure-State-Response Framework in Four Chinese Municipalities. Applied Sciences. 2022; 12:2819. https://doi.org/10.3390/app12062819

[7] Heshmati A, Rashidghalam M. Measurement and analysis of urban infrastructure and its effects on urbanization in China. Journal of Infrastructure Systems. 2020; 26: 04019030. https://doi.org/10.1061/(ASCE)IS.1943-555X.0000513

[8] Chen HL, Zhang Y, Zhang NX, Zhou M, Ding HP. Analysis on the Spatial Effect of Infrastructure Development on the Real Estate Price in the Yangtze River Delta. Sustainability. 2022; 14:7569. https://doi.org/10.3390/su14137569

[9] Tao ZM. Research on the degree of coupling between the urban public infrastructure system and the urban economic, social, and environmental system: A case study in Beijing, China. Mathematical Problems in Engineering. 2019; https://doi.org/10.1155/2019/8206902

……

[13] Zhang X, Zhong SQ, Ling S, Jia N, Qi H, He ZB. How to promote the transition from solo driving to mobility services delivery? An empirical study focusing on ridesharing. Transport Policy. 2022; 129: 176-187. https://doi.org/10.1016/j.tranpol.2022.10.009

2. Another aspect that the authors should improve on is to make the paper with more international and theoretical relevance, the main issue is the theoretical logic of resilience theory and urban infrastructure. 

Response: Thank you for your detailed suggestions. The assessment based on resilience theory includes three dimensions[39]: capacity, process, and goal. The capacity assessment emphasizes the ability and level of response in all areas of the city, the process assessment emphasizes the external disturbance processes that affect the dynamic improvement of the city's capacity, and the goal assessment emphasizes the achievement of the goal level after the city experiences a disaster. Urban infrastructure resilience emphasizes the existing state and capacity of each urban infrastructure subsystem to cope with disturbances, and its connotation fits with the capacity assessment dimension in resilience theory. Therefore, we combined the theoretical logic of resilience theory and urban infrastructure according to your proposal. Based on the existing implications, the international and theoretical implications of this study are further enriched in terms of both the capacity assessment dimension of resilience theory and its application in the field of urban infrastructure. The specific modifications are as follows (Page 3, lines 110-116):

“Based on the capacity assessment dimension of resilience theory [39], this study assesses the level of urban infrastructure resilience from the perspective of the existing capacity of urban infrastructure subsystems and explores the dynamic evolution characteristics and spatial spillover effects of urban infrastructure resilience. It helps to broaden the application field of resilience theory and expand the research horizon in the field of urban infrastructure, and it can provide references for other countries, particularly developing countries, to promote coordinated regional development, guide the construction of resilient cities, and reasonably avoid the urban siphon effect.” 

The references are as follows:

[39] Davidson-Hunt, I.J. Journeys, plants and dreams: Adaptive learning and social-ecological resilience. Ph.D. Thesis, The University of Manitoba, Winnipeg, MB, Canada, 2004.

3. As a specification manuscript, there are some shortcomings, there are some specific problems in the manuscript, such as the nine segment line display of China in Figure 3 and the failure to mark the Diaoyu Islands. 

Response: Thank you for your detailed suggestions. We apologize for the irregular graphics in the paper due to our oversight. Based on your suggestions, we have made targeted changes to Figure 3 and Figure 5 by adding the nine segment lines and labeling the Diaoyu Islands, among others.

4. What is the reason for the selection of the manuscripts for every decade after 2000? Policy context or particularly profound reasons in manuscripts. 

Response: Thank you for your suggestions. Due to our oversight, we failed to clearly describe the research period of the paper, and the research time points of this paper are 2010, 2015 and 2019. The reasons why we chose 2010 as the starting point of the study period are provided below. The research object of this study is 283 cities at the prefecture level and above in China, with a large sample size. With 2010 as the dividing year, the data for indicators such as the total social electricity consumption, the number of internet users per 10,000 people, and the number of health institutions are counted inconsistently. Due to objective factors such as inconsistency in statistical caliber, we chose 2010 as the starting point of the study period. Additionally, due to limitations such as an inconsistent statistical data updating speed and the novel coronavirus epidemic affecting data quality, we used 2019 as the cut-off year for this study.

5. Is the choice of indicator system scientific? Why is blue-green infrastructure not considered as an important element of urban resilience? 

Response: Thank you very much for your suggestions. Based on your suggestions, we explained the scientificity of the indicator system selection in the Construction of the Urban Infrastructure Resilience Index System section of the paper. The details are as follows (page 4-5, lines 157-172):

(1) Scientificity of the selection of the indicator system

This study considers the scientificity of the selection of the indicator system from the perspectives of whether these indicators comprehensively reflect the connotation of urban infrastructure resilience, whether the indicators covered by the existing relevant literature are comprehensively considered, and whether the data are available.

① Considerations from the connotation of infrastructure resilience

Infrastructure resilience emphasizes its state and ability to resist external perturbations. The 25 indicators used in this study, such as the gas penetration rate, total electricity consumption, water supply pipeline density, and drainage pipeline density, were selected because they are closely related to the connotation of infrastructure resilience.

② Consideration of infrastructure resilience indicators based on reference to the relevant literature

The current index system for urban infrastructure and its resilience is mainly constructed using energy, water supply and drainage, transportation, communication and other municipal utilities. For example, Chen et al. described the state of infrastructure resilience in terms of the gas penetration rate, water penetration rate, and road area per capita[6]. Liu, Chen, Tao, Bai, and Zhang et al. constructed indicator systems including daily domestic water consumption per capita, water supply pipeline density, drainage pipeline density, and road area per capita[5,7-13,26,27,40-42].

Using the principles of comprehensiveness, importance, measurability, and availability of evaluation indicators, we used the “Standard for Basic Terminology of Urban Planning” (GB/T50280-98) as the basis to count the frequency of indicators involved in 32 highly relevant papers, among which 21 indicators have a high usage rate (as shown in Table 1), and all these high-frequency indicators were included in the 25 indicators used in this study. Thus, we have selected four indicators, namely, special vehicles for sanitation for 10000 people, the number of secondary schools, the number of primary schools, and library collections per 100 people, as useful supplements to the engineering facilities and social service facilities in the connotation of infrastructure together to ensure that the index system is more comprehensive.

③ Considerations based on data availability

The research object of this study is 283 cities at the prefecture level and above in China, the sample size is large, and the data on indicators such as per capita shelter space area and emergency personnel ratio are not available. Therefore, after considering the comprehensiveness and availability of data, 25 representative indicators with a high utilization rate were identified to construct the urban infrastructure resilience evaluation system.

From the above analysis of the scientificity of the selection of the indicator system in this study, we can see that the indicators selected in this study are closely focused on the connotation of urban infrastructure resilience, covering the high-frequency indicators involved in the indicator system of related literature, and also taking into account the availability of data, etc., which has been in line with the scientific principle of the selection of indicators to the greatest extent. We believe that the choice of the indicator system is scientific.

Table 1. Comprehensive evaluation index system of urban infrastructure resilience.

Index Property Frequency Weight

Gas penetration rate Positive 32 0.043

Total gas supply Positive 28 0.026

Total electricity consumption Positive 25 0.040

Water supply pipeline density Positive 32 0.057

Water penetration rate Positive 28 0.037

Daily domestic water consumption per capita Negative 30 0.043

Drainage pipeline density Positive 32 0.047

Road area per capita Positive 32 0.048

Road network density Positive 29 0.047

Number of buses per 10,000 people Positive 31 0.032

Total number of bus passengers Positive 28 0.039

Number of households with internet access per 10,000 people Positive 30 0.061

Number of households with cell phones per 10,000 people Positive 32 0.033

Telecommunications services per capita Positive 29 0.035

Postal business per capita Positive 28 0.036

Centralized urban sewage treatment rate Positive 25 0.036

Harmless disposal rate of domestic waste Positive 26 0.020

Special vehicles for sanitation for 10,000 people Positive 10 0.030

Green space per capita Positive 29 0.047

Greening coverage of built-up areas Positive 25 0.044

Number of secondary schools Positive 5 0.040

Number of primary schools Positive 4 0.049

Library collections per 100 people Positive 10 0.042

Number of hospital beds per 1,000 population Positive 26 0.036

Number of health institutions Positive 29 0.032

(2) Considerations of the failure to consider blue‒green infrastructure as an important factor in urban resilience

Urban blue‒green infrastructure, including water systems, greenways, wetlands and forests, is an important factor in constructing the urban living environment, guiding the spatial layout of the city and enhancing urban resilience. The object of our study is mainly the municipal infrastructure of the city, with a greater emphasis on its macroscopic and holistic nature, which to some extent can include urban blue-green infrastructure. At the same time, considering the large differences in natural, economic and social conditions among prefecture-level cities in China and the lack of complete data on indicators related to urban blue‒green infrastructure. Therefore, we did not consider blue-green infrastructure separately as an important element of urban resilience in this study.

“The current index system for urban infrastructure and its resilience is mainly constructed in terms of energy, water supply and drainage, transportation, communication and other municipal utilities. For example, Chen et al. described the state of infrastructure resilience in terms of the gas penetration rate, water penetration rate, and road area per capita[6]. Liu, Chen, Tao, Bai, and Zhang et al. constructed indicator systems including the daily domestic water consumption per capita, water supply pipeline density, drainage pipeline density, and road area per capita [5,7-13,26,27,40-42].

This study follows the principles of comprehensiveness, importance, measurability, and availability of evaluation indicators and uses the “Standard for Basic Terminology of Urban Planning” (GB/T50280-98) as the basis for selection. The frequency of the indicators involved in 32 highly relevant papers is counted, and 21 indicators with representative and high usage are selected. Thus, we have selected four indicators, namely, special vehicles for sanitation for 10000 people, the number of secondary schools, the number of elementary schools, and library collections per 100 people, as useful supplements to the engineering facilities and social service facilities in the connotation of infrastructure together to ensure that the index system is more comprehensive. The finalized urban infrastructure resilience evaluation index system consists of 25 indicators such as the gas penetration rate, total gas supply, total electricity consumption, and water supply pipeline density (Table 1).”

Table 1. Comprehensive evaluation index system of urban infrastructure resilience.

Sub-index Index Property Frequency Weight

Resilience of energy facilities Gas penetration rate Positive 32 0.043

 Total gas supply Positive 28 0.026

 Total electricity consumption Positive 25 0.040

Resilience of supply and drainage facilities Water supply pipeline density Positive 32 0.057

 Water penetration rate Positive 28 0.037

 Daily domestic water consumption per capita Negative 30 0.043

 Drainage pipeline density Positive 32 0.047

Resilience of transport facilities Road area per capita Positive 32 0.048

 Road network density Positive 29 0.047

 Number of buses per 10,000 people Positive 31 0.032

 Total number of bus passengers Positive 28 0.039

Resilience of postal and communication facilities Number of households with internet access per 10,000 people Positive 30 0.061

 Number of households with cell phones per 10,000 people Positive 32 0.033

 Telecommunications services per capita Positive 29 0.035

 Postal business per capita Positive 28 0.036

Resilience of environmental facilities Centralized urban sewage treatment rate Positive 25 0.036

 Harmless disposal rate of domestic waste Positive 26 0.020

 Special vehicles for sanitation for 10,000 people Positive 10 0.030

 Green space per capita Positive 29 0.047

 Greening coverage of built-up areas Positive 25 0.044

Resilience of cultural and health facilities Number of secondary schools Positive 5 0.040

 Number of primary schools Positive 4 0.049

 Library collections per 100 people Positive 10 0.042

 Number of hospital beds per 1,000 population Positive 26 0.036

 Number of health institutions Positive 29 0.032

6. It is recommended to add some references, for example, “Spatial-Temporal Patterns of Network Structure of Human Settlements Competitiveness in Resource-Based Urban Agglomerations”，“Spatial Responses of Ecosystem Service Value during the Development of Urban Agglomerations” and “Morphological and functional polycentric structure assessment of megacity: An integrated approach with spatial distribution and interaction”. 

Response: Thank you very much for your suggestions. We have carefully studied your recommended references and read more recent related studies and have provided helpful additions to the literature in terms of the internal and external boundaries of urban infrastructure, the theoretical logic of resilience theory and urban infrastructure, factors influencing spatial spillover effects, and the important role of blue‒green infrastructure in urban resilience. Specific references include the 3rd, 4th, 5th, 7th, 8th, 9th, 10th, 11th, 12th, 13th, 30th, 31st, 39th, 51st, 55th, 58th, 59th, 60th, 61th, 62nd. There are 62 references now.

Once again, thank you very much for your constructive comments and suggestions, which have helped us a lot to improve the quality of the paper.

---

## [Decision Letter · Decision Letter 1]

30 Jan 2023

PONE-D-22-30335R1Dynamic evolution of urban infrastructure resilience and its spatial spillover effects: An empirical study from ChinaPLOS ONE

Dear Dr. Liang,

Thank you for submitting your manuscript to PLOS ONE. After careful consideration, we feel that it has merit but does not fully meet PLOS ONE’s publication criteria as it currently stands. Therefore, we invite you to submit a revised version of the manuscript that addresses the points raised during the review process.

We look forward to receiving your revised manuscript.

Kind regards,

Jun Yang

Academic Editor

PLOS ONE

Journal Requirements:

Additional Editor Comments:

Thanks for your replies - please note the numbers below indicate the order of authors' responses to my previous questions. Q1-OK Q2-the authors highlighted the innovation, it should be further improved. Q3-OK Q4-OK Q5-OK

Reviewers' comments:

Reviewer's Responses to Questions

**Comments to the Author**

1. If the authors have adequately addressed your comments raised in a previous round of review and you feel that this manuscript is now acceptable for publication, you may indicate that here to bypass the “Comments to the Author” section, enter your conflict of interest statement in the “Confidential to Editor” section, and submit your "Accept" recommendation.

Reviewer #1: All comments have been addressed

Reviewer #2: (No Response)

2. Is the manuscript technically sound, and do the data support the conclusions?

Reviewer #1: Yes

Reviewer #2: (No Response)

3. Has the statistical analysis been performed appropriately and rigorously? 

Reviewer #1: Yes

Reviewer #2: (No Response)

4. Have the authors made all data underlying the findings in their manuscript fully available?

Reviewer #1: Yes

Reviewer #2: (No Response)

5. Is the manuscript presented in an intelligible fashion and written in standard English?

Reviewer #1: Yes

Reviewer #2: (No Response)

6. Review Comments to the Author

Reviewer #1: Thanks for your replies - please note the numbers below indicate the order of authors' responses to my previous questions.

Q1-OK

Q2-the authors highlighted the innovation, it should be further improved.

Q3-OK

Q4-OK

Q5-OK

Reviewer #2: (No Response)

7. PLOS authors have the option to publish the peer review history of their article (what does this mean?). If published, this will include your full peer review and any attached files.

Reviewer #1: No

Reviewer #2: No

---

## [Author Response · Author response to Decision Letter 1]

5 Feb 2023

Reply to-comments

Manuscript Number: PONE-D-22-30335

Summary of the major changes and responses to the Reviewers’ comments: 

Dear Editor,

On behalf of my co-authors, we thank you very much for giving us an opportunity to revise our manuscript again. We greatly appreciate the editors for their positive constructive comments and suggestions on our manuscript entitled “Dynamic evolution of urban infrastructure resilience and its spatial spillover effects: An empirical study from China”. (ID: PONE-D-22-30335).

We have studied the editors’ comments carefully and addressed them one by one. Attached please find the response to the editor and revised manuscript, which we would like to submit for your kind consideration. All modifications in the manuscript are indicated in red font.

We would like to express our great appreciation to you for your comments on our paper. We look forward to hearing from you.

Thank you and best regards.

Yours sincerely,

Hao Wang

School of Geographic Sciences, Hebei Normal University, Shijiazhuang 050024, China

Tel: 86-031180787632; E-mail: liangyanqing@126.com

 

Responses to Anonymous Reviewer #1’s Comments:

Specific comments:

Thanks for your replies - please note the numbers below indicate the order of authors' responses to my previous questions. Q1-OK Q2-the authors highlighted the innovation, it should be further improved. Q3-OK Q4-OK Q5-OK.

Response: On behalf of my coauthors, we greatly appreciate your positive and constructive comments and suggestions on our manuscript. We apologize for insufficiently improving the innovations of the article. In accordance with your suggestions, we have further emphasized the points of innovation of the paper, which mainly focus on three aspects: the indicator system, research perspective, and research scale. The specific information is as follows (page 3, lines 109-116):

“In summary, first, this paper constructs a comprehensive UIR evaluation system by integrating energy, supply and drainage, transport, postal and communication, environmental, and cultural and health facilities. Second, this paper explores the spatial spillover effects and causes of the variability in infrastructure resilience among cities at different time points using a spatial econometric model that considers spatial dependence. Again, this paper takes 283 cities at the prefecture level and above in China as its research objects to more intuitively reflect the spatial heterogeneity of UIR levels at the municipal scale, which is an important socioeconomic management unit.”

Once again, thank you very much for your constructive comments and suggestions, which have helped us a lot to improve the quality of the paper.

---

## [Decision Letter · Decision Letter 2]

10 Feb 2023

Dynamic evolution of urban infrastructure resilience and its spatial spillover effects: An empirical study from China

PONE-D-22-30335R2

Dear Dr. Liang,

We’re pleased to inform you that your manuscript has been judged scientifically suitable for publication and will be formally accepted for publication once it meets all outstanding technical requirements.

Kind regards,

Jun Yang

Academic Editor

PLOS ONE

Additional Editor Comments (optional):

Accept

Reviewers' comments:

Reviewer's Responses to Questions

**Comments to the Author**

1. If the authors have adequately addressed your comments raised in a previous round of review and you feel that this manuscript is now acceptable for publication, you may indicate that here to bypass the “Comments to the Author” section, enter your conflict of interest statement in the “Confidential to Editor” section, and submit your "Accept" recommendation.

Reviewer #1: (No Response)

Reviewer #2: (No Response)

2. Is the manuscript technically sound, and do the data support the conclusions?

Reviewer #1: (No Response)

Reviewer #2: (No Response)

3. Has the statistical analysis been performed appropriately and rigorously? 

Reviewer #1: (No Response)

Reviewer #2: (No Response)

4. Have the authors made all data underlying the findings in their manuscript fully available?

Reviewer #1: (No Response)

Reviewer #2: (No Response)

5. Is the manuscript presented in an intelligible fashion and written in standard English?

Reviewer #1: (No Response)

Reviewer #2: (No Response)

6. Review Comments to the Author

Reviewer #1: (No Response)

Reviewer #2: (No Response)

7. PLOS authors have the option to publish the peer review history of their article (what does this mean?). If published, this will include your full peer review and any attached files.

Reviewer #1: No

Reviewer #2: No

---

## [Editor Report · Acceptance letter]

6 Mar 2023

PONE-D-22-30335R2 

Dynamic evolution of urban infrastructure resilience and its spatial spillover effects: An empirical study from China 

Dear Dr. Liang:

I'm pleased to inform you that your manuscript has been deemed suitable for publication in PLOS ONE. Congratulations! Your manuscript is now with our production department. 

Kind regards, 

on behalf of

Dr. Jun Yang 

Academic Editor

PLOS ONE